# O-GlcNAcylation regulates neurofilament-light assembly and function and is perturbed by Charcot-Marie-Tooth disease mutations

Duc T. Huynh [1], Kalina N. Tsolova[1], Abigail J. Watson [1], Sai Kwan Khal [1], Jordan R. Green [2], Di Li [1], Jimin Hu[1], Erik J. Soderblom[3], Jen-Tsan Chi [4], Chantell S. Evans[2] & Michael Boyce [1,2] ✉

The neurofilament (NF) cytoskeleton is critical for neuronal morphology and function. In particular, the neurofilament-light (NF-L) subunit is required for NF assembly in vivo and is mutated in subtypes of Charcot-Marie-Tooth (CMT) disease. NFs are highly dynamic, and the regulation of NF assembly state is incompletely understood. Here, we demonstrate that human NF-L is modified in a nutrient-sensitive manner by O-linked-β-N-acetylglucosamine (O-GlcNAc), a ubiquitous form of intracellular glycosylation. We identify five NF-L O-GlcNAc sites and show that they regulate NF assembly state. NF-L engages in O-GlcNAc-mediated protein-protein interactions with itself and with the NF component α-internexin, implying that O-GlcNAc may be a general regulator of NF architecture. We further show that NF-L O-GlcNAcylation is required for normal organelle trafficking in primary neurons. Finally, several CMT-causative NF-L mutants exhibit perturbed O-GlcNAc levels and resist the effects of O-GlcNAcylation on NF assembly state, suggesting a potential link between dysregulated O-GlcNAcylation and pathological NF aggregation. Our results demonstrate that site-specific glycosylation regulates NF-L assembly and function, and aberrant NF O-GlcNAcylation may contribute to CMT and other neurodegenerative disorders.

The unique morphology, homeostasis, and functions of neurons depend on a dynamic and highly regulated cytoskeleton, comprising actin, microtubule, and intermediate filament (IF) compartments[1]. Neurofilaments (NFs) are neuronal IFs, composed of three major "triplet" subunits (light, medium, and heavy; NF-L, NF-M, and NF-H) that form heterotypic polymers with each other and with two additional NF proteins, α-internexin (INA) and peripherin, in the central and peripheral nervous systems, respectively[2]. All NF proteins contain an α-helical coiled-coil rod domain flanked by amino-terminal head and carboxyl-terminal tail domains of varying lengths[2,3]. NF proteins assemble in discrete states, with two monomers first forming coiled-coils in the rod domain to create head-to-head dimers[2,3]. Two dimers then assemble in antiparallel fashion, forming a nonpolar tetramer[2,3]. Eight tetramers anneal laterally to form ~65 nm unit-length filaments, which elongate end-to-end into short filaments and then compact radially into mature, fully assembled NFs of ~10 nm diameter[2,3]. Due to their nonpolar nature, IFs cannot serve as tracks for molecular motors but instead exhibit viscoelastic properties distinct from the actin or microtubule networks[1,4–6]. For example, IFs are flexible under low strain but rigidify and resist breakage under applied force, with fully assembled filaments stiffening more than lower-order oligomers[1,3–6].

[1]Department of Biochemistry, Duke University School of Medicine, Durham, NC 27710, USA. [2]Department of Cell Biology, Duke University School of Medicine, Durham, NC 27710, USA. [3]Proteomics and Metabolomics Shared Resource, Duke University School of Medicine, Durham, NC 27710, USA. [4]Department of Molecular Genetics and Microbiology, Duke University School of Medicine, Durham, NC 27710, USA. ✉e-mail: michael.boyce@duke.edu

Therefore, the assembly state of all IFs, including NFs, is critical for their contributions to cell physiology.

Of the triplet proteins, NF-L is required for the structural integrity of axons[7–10], has been detected at post-synaptic sites[11], and directly interacts with the *N*-methyl-ᴅ-aspartate receptor to influence high-order brain functions[12]. Ablating the *NEFL* gene, which encodes NF-L, impairs the maturation of regenerating myelinated axons[7], dendritic arborization[8], and peripheral nerve regeneration in mice[9]. Motor neurons derived from human *NEFL*[−/−] induced pluripotent stem cells (iPSC) show reduced axonal caliber, dysregulated mitochondrial motility, and decreased electrophysiological activity[10]. Moreover, NF-L assembly state and functions are perturbed in a range of nervous system disorders. *NEFL* mutations cause some subtypes of Charcot–Marie–Tooth (CMT) disease[13], an inherited peripheral neuropathy characterized by progressive atrophy of the distal limb muscles that leads to sensory loss and tendon reflex defects[14]. CMT-causative mutations in *NEFL* or other genes result in NF aggregation and aberrant motility of neuronal mitochondria[15–21], underlining the physiological importance of NF-L function. Aggregation of wild-type (WT) NF-L and other NF proteins is also a pathological hallmark of a variety of neurological conditions, including Alzheimer's disease (AD)[22], Parkinson's disease (PD)[23], amyotrophic lateral sclerosis (ALS)[24], giant axonal neuropathy (GAN)[16], and spinal muscular atrophy[25]. Recently, NF-L levels in the cerebrospinal fluid (CSF) and blood have emerged as powerful biomarkers of neuronal injury and nervous system disorders[2,26], showing promise for early diagnosis in a variety of clinical settings[27–29]. Given this broad pathophysiological significance, elucidating the regulation of NF assembly state and functions is a key goal. However, the molecular mechanisms governing NFs remain incompletely understood.

One major mode of NF regulation is likely through various post-translational modifications (PTMs)[2,30]. Prior proteomic studies indicated that rodent NFs are modified by O-linked-β-*N*-acetylglucosamine (O-GlcNAc)[31–34], an abundant intracellular form of glycosylation reversibly decorating serine and threonine sidechains on many nuclear, cytoplasmic, and mitochondrial proteins (Fig. 1a)[35]. In mammals, O-GlcNAc is added by O-GlcNAc transferase (OGT) and removed by O-GlcNAcase (OGA), both ubiquitous nucleocytoplasmic enzymes[35]. O-GlcNAc cycling is essential, as deletion of *OGT* or *OGA* is lethal in mice[36,37]. O-GlcNAc occurs in nearly all mammalian tissue types[35] and is especially prevalent in the brain and post-synaptic densities[38,39]. Ablating the *OGT* gene in specific populations of murine dopaminergic[40], hypothalamic[41], or cerebellar[42] neurons causes cellular and behavioral defects, demonstrating the importance of O-GlcNAc to brain function. At the molecular level, O-GlcNAc mediates various aspects of mammalian neuronal biology[43], and O-GlcNAcylation of disease-relevant substrates, such as tau[44] or α-synuclein[45], is dysregulated in multiple clinically important neurological disorders[46]. Manipulating O-GlcNAc in the nervous system has shown therapeutic promise in recent pre-clinical studies[47–51] and clinical trials[52–54] alike. For example, multiple reports showed that elevating O−GlcNAcylation via treatment with the small molecule OGA inhibitor Thiamet-G[47] reduced proteotoxicity, cognitive deficits, and behavioral dysfunction in AD rodent models[48,49]. Based on these and other promising pre-clinical data[47,50–54], at least three OGA inhibitors (MK-8719, LY3372689, ASN90) have entered human clinical trials[52–54], paving the way for future pharmacological modulation of O-GlcNAc on substrates, such as NF proteins, in human patients. However, despite its potential clinical significance, the functional impact of O-GlcNAcylation on human NF proteins has never been studied systematically.

Here, we demonstrate that human NF-L is O-GlcNAc-modified in cell culture models, primary neurons, and post-mortem brain tissue. NF-L assembly state and function are regulated by site-specific O-GlcNAcylation, as elevating O-GlcNAc drives NF-L to lower-order oligomeric states, reduces the prevalence of full-length filaments, and alters organelle motility in primary hippocampal neurons. At the molecular level, we show that NF-L O-GlcNAcylation is nutrient-responsive and mediates both homotypic NF-L/NF-L and heterotypic NF-L/INA interactions, revealing glycan-mediated interactions among NF components. Further, we observed aberrant O-GlcNAc levels on CMT-causative NF-L mutants and hypoglycosylation of NF-L when CMT mutations lie proximal to glycosites. Hypoglycosylated CMT NF-L mutants formed aggregates that were insensitive to the normal assembly state effects of O-GlcNAcylation, compared to WT NF-L. Together, our results indicate that site-specific O-GlcNAcylation is a mode of NF regulation and may be dysregulated in neurological disorders.

## Results

### Site-specific O-GlcNAcylation of the human NF-L head and tail domains

Several prior studies reported the O-GlcNAcylation of rodent NF proteins[31–34]. However, the existence and functions of O-GlcNAc on human NFs had not been examined systematically. To address this knowledge gap, we first focused on NF-L because it is essential for NF assembly in vivo[2] and is mutated in subtypes of CMT disease[13]. In cells expressing tagged human NF-L, OGT co-expression induced global O-GlcNAcylation and elevated NF-L O-GlcNAc levels, as judged by immunoprecipitation (IP) experiments that solubilize the vast majority of NF-L (Fig. 1b). In contrast, expression of OGT[H498A], a glycosyltransferase-dead mutant[55], did not affect NF-L O-GlcNAcylation, confirming the specificity of the assay (Supplementary Fig. 1A). Inhibiting OGA or OGT with the small molecules Thiamet-G or peracetylated 5-thio-GlcNAc (5SGlcNAc)[56] elevated or reduced NF-L O-GlcNAcylation, respectively (Fig. 1c). In human neuroblastoma SH-SY5Y cells, Thiamet-G treatment increased endogenous NF-L O-GlcNAcylation in a time-dependent manner (Fig. 1d). Furthermore, we detected endogenous NF-L O-GlcNAcylation in post-mortem human temporal cortex, frontal cortex, and parietal cortex tissue samples (Fig. 1e and Supplementary Fig. 1B). Mild β-elimination of O-linked glycans[57] extinguished the signal on these anti-O-GlcNAc immunoblots (IBs), confirming it as authentic (Supplementary Fig. 1C). These results demonstrate that human NF-L is dynamically O-GlcNAcylated in culture models and in vivo.

Previous reports identified several O-GlcNAcylation sites on rodent NF-L orthologs[31–34], but directed studies of human NF glycosites were lacking. To discover human NF-L glycosites, we epitope-tagged the *NEFL* genomic locus of human cells via established CRISPR/Cas9 methods[58] (Supplementary Fig. 1D) and affinity-purified endogenous NF-L from cultures treated with Thiamet-G, a standard tactic to improve the technically challenging detection of O-GlcNAc moieties[59,60]. Mass spectrometry (MS) analysis identified two novel glycosites (S48, S431), complementing the previously reported glycosites from rodent NF-L[31,32] (T21, S27, S34−human numbering of cognate rodent residues) (Fig. 1f). Next, we created an unglycosylatable S/T→A mutation at each of these residues and measured the effects on total NF-L O-GlcNAcylation by IP and quantitative fluorescent IB to determine major glycosites. While all NF-L constructs displayed comparable basal levels of O-GlcNAcylation (Supplementary Fig. 1E), each mutant except T21A exhibited lower OGT-induced O-GlcNAcylation than WT (Fig. 1g, h), suggesting that NF-L could be simultaneously O-GlcNAcylated at multiple residues in response to upstream stimuli. Consistent with this hypothesis, an NF-L mutant with all four head-domain glycosites changed to alanine (NF-L[4A]) exhibited significantly less induced O-GlcNAcylation, compared to WT (Fig. 1h, i). The single S431A mutation in the tail domain glycosite also reduced NF-L O-GlcNAcylation, whereas a compound mutant lacking the four head and one tail domain sites (NF-L[5A]) exhibited total O-GlcNAcylation similar to the NF-L[4A] mutant (Fig. 1h, j), suggesting potential inter-domain interactions during OGT modification. Taken together, these

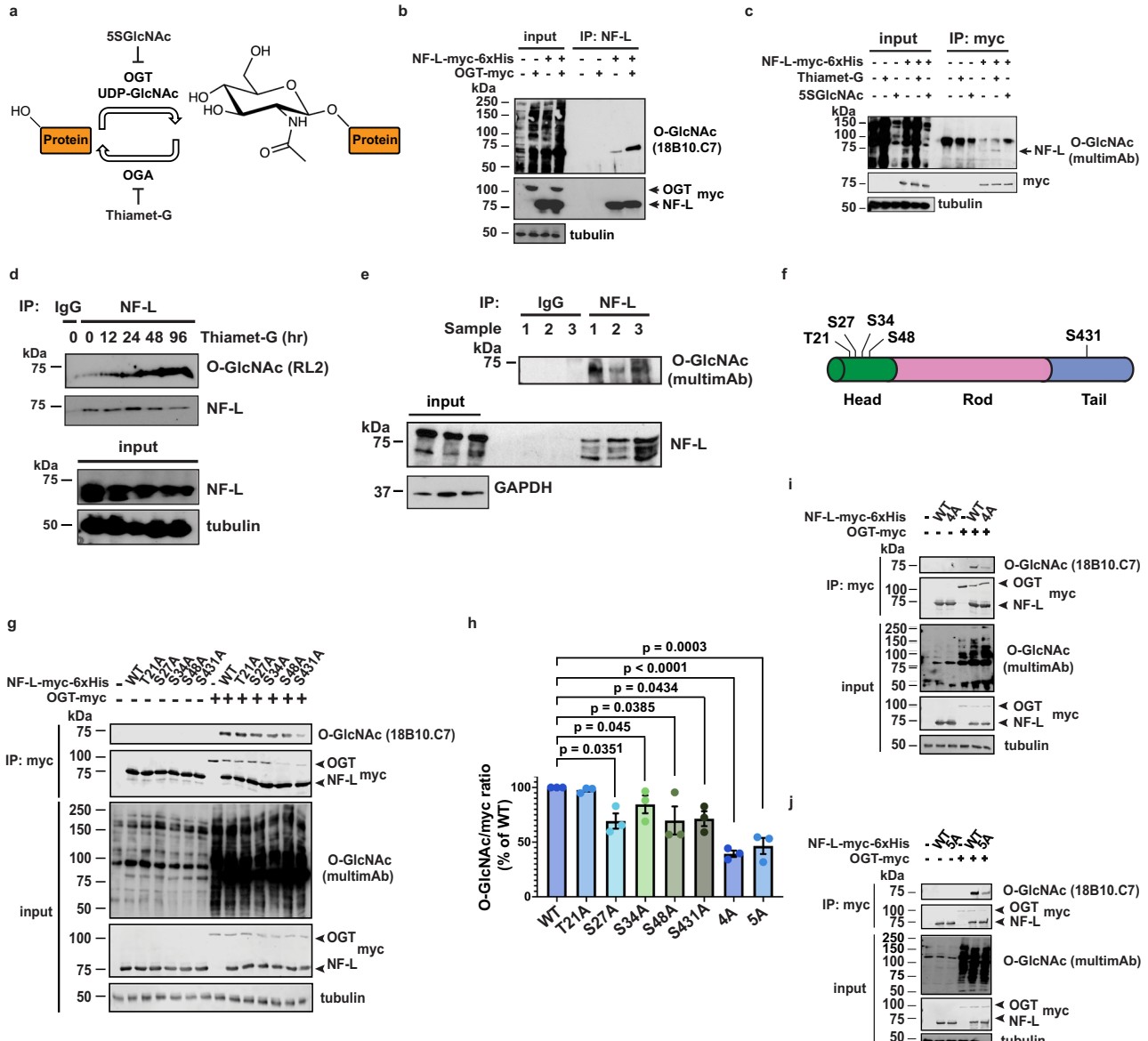

**Fig. 1 | Site-specific O-GlcNAcylation of the human NF-L head and tail domains.**
**a** O-GlcNAc transferase (OGT) uses the nucleotide-sugar UDP-GlcNAc to add
O-GlcNAc to serine or threonine residues of intracellular proteins, and O-GlcNAcase
(OGA) catalyzes its removal. OGT and OGA can be inhibited by the small molecules
5SGlcNAc and Thiamet-G, respectively. **b** 293T cells were transfected with NF-L-
myc-6xHis ± OGT-myc for 24 h, and lysates were analyzed by NF-L IP and IB ($n$ = 3
biological replicates). **c** 293T cells were transfected with NF-L-myc-6xHis for 24 h,
treated with 50 μM Thiamet-G or 50 μM 5SGlcNAc for 30 h, and lysates were ana-
lyzed by myc IP and IB ($n$ = 3 biological replicates). **d** SH-SY5Y cells were cultured in
50 μM Thiamet-G for the times indicated, and lysates were analyzed by control

(IgG) or NF-L IP and IB ($n$ = 2 biological replicates). **e** Human temporal cortex
homogenates from three donors (1–3) were analyzed by control (IgG) or NF-L IP and
IB. **f** Five O-GlcNAc sites identified by MS in this work and prior studies are indicated
on the NF-L domain structure. **g** $NEFL^{-/-}$ 293T cells were transfected with NF-L-myc-
6xHis WT or single-point glycosite mutants ± OGT-myc for 24 h, and lysates were
analyzed by myc IP and IB. **h** Normalized O-GlcNAc signal (O-GlcNAc/myc ratio) was
calculated for the experiments performed in (**g, i, j**). Data are shown as mean ± SEM
and assessed by one-way ANOVA/Tukey's post hoc correction ($n$ = 3 biological
replicates). **i, j** $NEFL^{-/-}$ 293T cells were transfected with NF-L-myc-6xHis WT or NF-
L$^{4A}$ (**i**) or NF-L$^{5A}$ (**j**) ± OGT-myc for 24 h, and lysates were analyzed by myc IP and IB.

data demonstrate the inducible, site-specific O-GlcNAcylation of
human NF-L.

O-GlcNAcylation and phosphorylation engage in well-
documented signaling crosstalk, sometimes competing for identical
or nearby residues on the same substrates[33,61–65]. None of the NF-L O-
GlcNAc sites we identified is a validated phosphosite, but NF-L is
known to be phosphorylated at other residues[66,67]. Therefore, we tes-
ted the potential relationship between these PTMs on NF-L. OGT co-
expression caused a modest reduction in NF-L phosphorylation (Sup-
plementary Fig. 1F), whereas augmenting global phosphorylation via
treatment with calyculin A, a potent inhibitor of the serine/threonine
phosphatases PP1 and PP2A, slightly decreased in NF-L O-

GlcNAcylation (Supplementary Fig. 1G). In addition, baseline phos-
phorylation of the NF-L$^{4A}$ and NF-L$^{5A}$ mutants was modestly elevated,
compared to WT (Supplementary Fig. 1E). These results suggest that
O-GlcNAcylation and phosphorylation may be reciprocal PTMs of NF-
L, though future studies will be needed to identify the relevant phos-
phorylation site(s) and the potential downstream functional implica-
tions of this interplay.

### NF-L O-GlcNAcylation influences assembly state and filament formation

The NF-L head domain is required for its assembly[2], and other PTMs,
including phosphorylation, on distinct head domain residues regulate

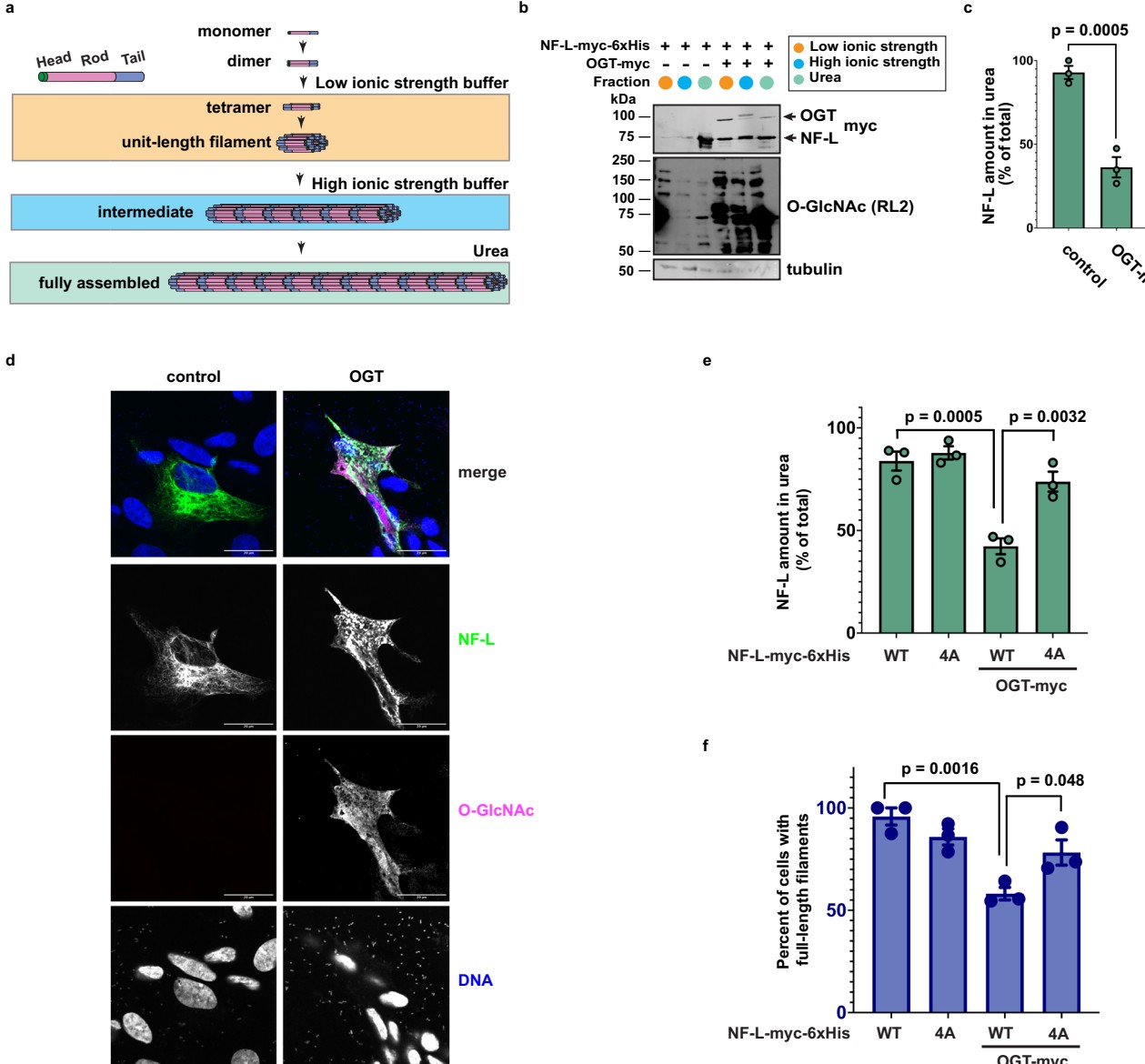

**Fig. 2 | NF-L O-GlcNAcylation influences NF-L assembly state and filament formation. a** Differential extraction assay separates discrete assembly states of NF-L. Low ionic strength, low-order assembly states; high ionic strength, intermediate assembly states; 8 M urea, fully assembled NFs. **b** 293T cells were transfected with NF-L-myc-6xHis ± OGT-myc for 24 h and analyzed by differential extraction and IB (*n* = 3 biological replicates). **c** NF-L amount extracted into urea buffer was calculated as percent of total NF-L across three fractions from the experiment described in (**b**). Data are shown as mean ± SEM and assessed by Student's two-tailed *t* test (*n* = 3 biological replicates). **d** *NEFL*⁻/⁻ SH-SY5Y cells were transfected with NF-L-myc-6xHis ± OGT-myc for 24 h and analyzed by IFA (*n* = 3 biological replicates).

Scale bar: 20 μm. **e** *NEFL*⁻/⁻ 293T cells were transfected with WT or NF-L⁴ᴬ-myc-6xHis ± OGT-myc for 24 h and analyzed by differential extraction and IB. NF-L amount extracted into urea buffer was calculated as percent of total NF-L across three fractions. Data are shown as mean ± SEM and assessed by one-way ANOVA/Tukey's post hoc correction (*n* = 3 biological replicates). **f** *NEFL*⁻/⁻ SH-SY5Y cells were transfected with WT or NF-L⁴ᴬ-myc-6xHis ± OGT-myc for 24 h and analyzed by IFA. Quantification of percent of cells with full-length NFs was performed by a blinded researcher. Data are shown as mean ± SEM and assessed by one-way ANOVA/Tukey's post hoc correction (*n* = 3 biological replicates).

this process[66,67]. Therefore, we examined whether head domain O-GlcNAcylation affects NF-L assembly state. We used an established differential extraction assay that biochemically enriches distinct IF assembly states (Fig. 2a)[68] to determine the effects of O-GlcNAcylation on the NF-L network. As expected, WT NF-L was extracted overwhelmingly into a denaturing urea buffer, indicating that it was assembled into full-length filaments, which are poorly soluble in the other buffers used (Fig. 2b, c). In multiple human cell types, OGT co-expression significantly increased the solubility of NF-L in non-denaturing buffers of low or high ionic strength (Fig. 2b, c and Supplementary Fig. 2A, B), indicating that increased global

O-GlcNAcylation drives NF-L to lower-order assembly states and reduces the prevalence of full-length filaments. In contrast, co-expression of OGTᴴ⁴⁹⁸ᴬ did not affect the NF-L extraction profile (Supplementary Fig. 2C), demonstrating a requirement for glycosyltransferase activity. To confirm these results via an independent approach, we turned to immunofluorescence assays (IFA). We re-expressed NF-L in *NEFL*⁻/⁻ SH-SY5Y cells, which retain endogenous expression of other NF proteins (Supplementary Fig. 2D–F), with or without OGT co-expression (Fig. 2d). Consistent with our differential extraction data, IFA showed that NF-L formed intact filaments when expressed alone, but both filaments and lower-order oligomeric states

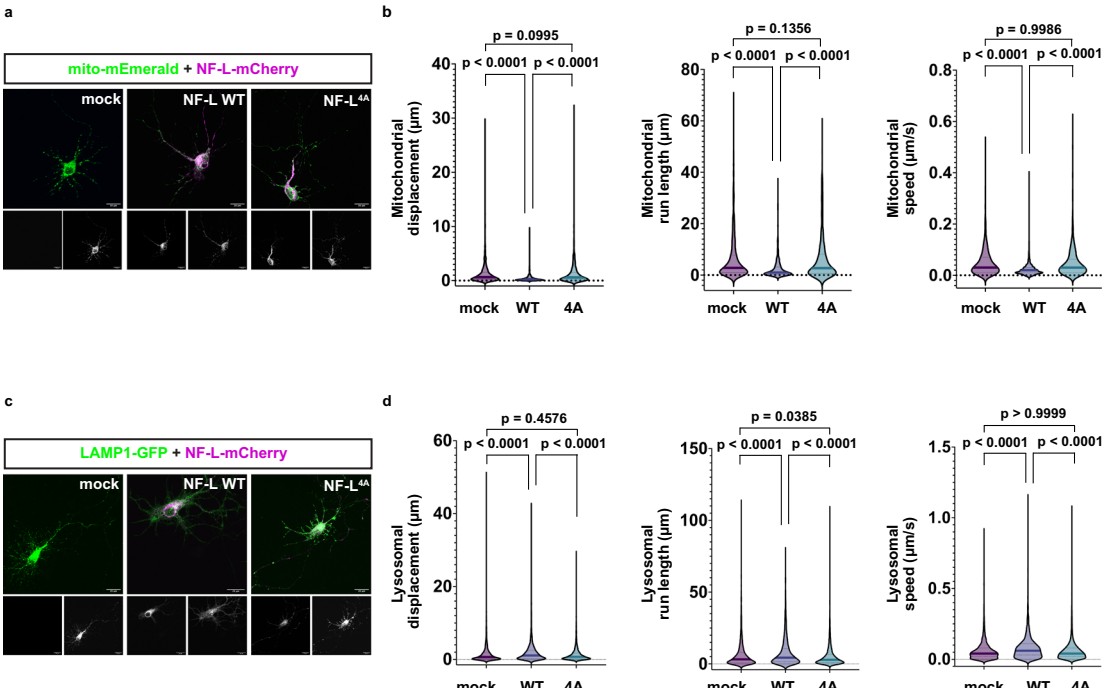

**Fig. 3 | NF-L head domain O-GlcNAcylation is required for organelle motility regulation. a** Cultured E18 rat hippocampal neurons at day 6 in vitro were transfected with WT or NF-L⁴ᴬ-mCherry + mito-mEmerald (mitochondrial marker) for 24 h and analyzed by live-cell fluorescence microscopy (n = 3 biological replicates). Scale bar: 20 μm. **b** Mitochondrial motility in axons of neurons treated as in (**a**) was imaged every 5 s for 5 min and analyzed to obtain organelle displacement, run length, and speed. Violin plots indicate median (bold line) and 25th and 75th percentiles (dotted lines), and data were assessed by Kruskal–Wallis test/Dunn's post hoc correction (n = 3 biological replicates). **c** Cultured rat hippocampal neurons at day 6 in vitro were transfected with WT or NF-L⁴ᴬ-mCherry + LAMP1-GFP (lysosomal marker) for 24 h (n = 3 biological replicates). Scale bar: 20 μm. **d** Lysosomal motility in axons of neurons treated as in (**c**) was imaged every 5 s for 5 min and analyzed to obtain displacement, run length, and speed. Violin plots indicate median (bold line) and 25th and 75th percentiles (dotted lines), and data were assessed by Kruskal–Wallis test/Dunn's post hoc correction (n = 3 biological replicates).

were observed upon OGT co-expression (Fig. 2d). These results demonstrate that increased global O-GlcNAc levels drive NF-L from a full-length filament assembly state to lower-order states.

OGT has thousands of substrates in human cells[35] and could impact NF-L assembly state through direct and/or indirect mechanisms. To determine whether O-GlcNAcylation of NF-L itself governs NF assembly state, we performed differential extraction assays on WT and glycosite mutants of NF-L expressed in a tractable *NEFL*⁻/⁻ human cell system (Supplementary Fig. 2G, H). Consistent with our observation of uniform baseline glycosylation across NF-L WT and mutants (Supplementary Fig. 1E), the individual glycosite mutants displayed no changes in extraction profile, compared to WT, when expressed alone (Supplementary Fig. 2I, J). In contrast, in the presence of OGT co-expression, NF-L⁴ᴬ resisted the shift to lower-order assembly states exhibited by WT protein (Fig. 2e), demonstrating that O-GlcNAcylation at specific sites in the NF-L head domain influences its assembly state. We obtained similar results in *NEFL*⁻/⁻ SH-SY5Y cells using an independent IFA assay: At baseline, NF-L⁴ᴬ displayed a reduced prevalence of full-length filaments, compared to WT, but OGT expression did not decrease the NF-L⁴ᴬ full-length filament population, as it does for WT (Fig. 2f).

In vivo, NF-L typically co-assembles with NF-M or NF-H to form mature NFs[19,69–72]. To assess the impact of NF-L O-GlcNAcylation on NF heteropolymer morphology, we co-expressed WT NF-L or NF-L⁴ᴬ with NF-M or NF-H in a physiological 4:2:1 ratio in SW13 vim⁻ cells, a tractable and well-established model system that lacks all cytoplasmic IFs[73] and is frequently used for imaging studies of IF protein mutations[19,72,74]. While WT NF-L co-assembled with NF-M or NF-H, forming full-length filaments as expected, the co-assembly of NF-L⁴ᴬ was impaired, exhibiting both full-length and shorter filaments (Supplementary Fig. 2K, L).

Taken together, these results demonstrate that site-specific O-GlcNAcylation in the NF-L head domain regulates NF assembly state.

## NF-L head domain O-GlcNAcylation is required for organelle motility regulation

The NF network regulates organelle motility, and loss of NF-L causes accelerated mitochondrial movement in several neuronal model systems[10,15]. Therefore, we used live-cell fluorescence microscopy[75] to test whether NF-L O-GlcNAcylation impacts organelle motility. Consistent with prior studies, expression of WT NF-L in cultured rat hippocampal neurons reduced mitochondrial total displacement, run length, and speed (Fig. 3a, b). By contrast, expression of NF-L⁴ᴬ did not significantly affect mitochondrial motility, as results from mock-transfected and NF-L⁴ᴬ-expressing neurons were statistically indistinguishable (Fig. 3b). Notably, expression of WT NF-L increased lysosomal motility (displacement, run length, speed), whereas NF-L⁴ᴬ expression did not affect these parameters (Fig. 3c, d). The NF network indirectly influences organelle motility by creating steric barriers and/or through interactions with the microtubule cytoskeleton[76], which forms the tracks for motor-driven organelle transport[77]. However, expression of WT NF-L or NF-L⁴ᴬ did not detectably affect the neuronal microtubule network (Supplementary Fig. 3A), ruling out a general cytoskeletal disruption caused by loss of NF-L glycosylation. These results indicate that NF-L head domain O-GlcNAcylation modulates organelle motility in primary neurons.

## Nutrient dependence of NF-L O-GlcNAcylation

Our organelle motility data indicate that basal levels of O-GlcNAcylation are required for some NF-L functions because WT NF-L and NF-L⁴ᴬ exhibited distinct phenotypes even in the absence of

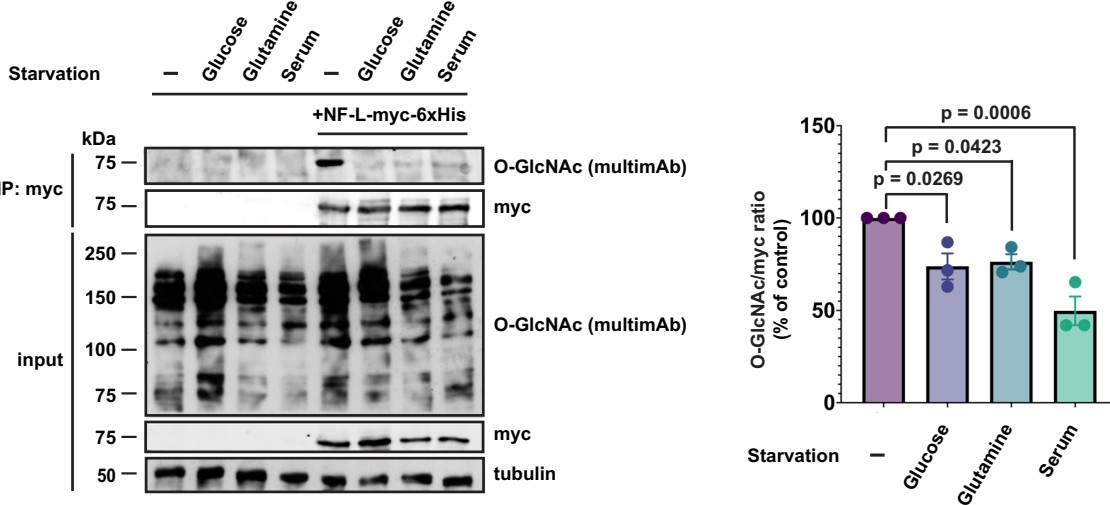

**Fig. 4 | Nutrient dependence of NF-L O-GlcNAcylation.** Left: *NEFL*[−/−] 293T cells were transfected with NF-L-myc-6xHis for 24 h, grown in control medium or medium without glucose, glutamine, or serum for 48 h, and analyzed by myc IP and

IB. Right: Normalized O-GlcNAc signal (O-GlcNAc/myc ratio) was calculated. Data are shown as mean ± SEM and assessed by one-way ANOVA/Tukey's post hoc correction (*n* = 3 biological replicates).

applied stimuli (Fig. 3). However, elevating global O-GlcNAc levels with an experimental stimulus (e.g., OGT expression) revealed other phenotypes that were not evident at baseline (Fig. 2 and Supplementary Fig. 2I, J), indicating that upstream signals might regulate NF-L by inducing or inhibiting its O-GlcNAcylation. One such candidate stimulus is a change in nutrient or growth factor availability. O-GlcNAc is well-known to act in part as a sensor of nutrients, such as glucose and glutamine, which are biosynthetic precursors of uridine diphosphate (UDP)-GlcNAc, the nucleotide-sugar cofactor used by OGT (Fig. 1a)[35]. Moreover, we have previously shown that fluctuations in nutrients or growth factors affect the O-GlcNAcylation of gigaxonin, a ubiquitin E3 ligase adaptor that targets NF proteins for proteasome-mediated destruction[59]. These results provided a precedent for metabolite-dependent regulation of NF-L through O-GlcNAc signaling. As a first step toward determining whether the O-GlcNAcylation of NF-L itself is affected by nutrient fluctuations, we starved cells of glucose, glutamine, or serum and quantified NF-L O-GlcNAcylation (Fig. 4). Interestingly, all three starvation treatments resulted in significant reductions in NF-L glycosylation but not NF-L expression (Fig. 4). These results establish nutrient and growth factor fluctuations as candidate stimuli that may impact NF-L O-GlcNAcylation and function in vivo.

## Direct, O-GlcNAc-mediated interactions between NF-L and INA

We next sought to define the molecular mechanism by which O-GlcNAc influences NF-L assembly state. O-GlcNAc can mediate protein-protein interactions (PPIs) on a range of substrates[35], and we have previously shown that the IF protein vimentin engages in homotypic, O-GlcNAc-mediated PPIs that are essential for filament formation[68]. Since NF-L both self-associates and co-polymerizes with other NF proteins into higher-order complexes[2], we hypothesized that NF-L O-GlcNAcylation might influence NF assembly states by mediating PPIs. However, physiological O-GlcNAc-mediated interactions are often low-affinity and sub-stoichiometric, making them technically challenging to characterize[35]. To overcome this obstacle, we employed a chemical biology method to capture endogenous O-GlcNAc-mediated PPIs[78]. Briefly, live cells are first treated with Ac$_3$GlcNDAz-1P(Ac-SATE)$_2$, a precursor form of "GlcNDAz," a GlcNAc analog that bears a diazirine photocross-linking moiety. The Ac$_3$GlcNDAz-1P(Ac-SATE)$_2$ precursor is peracetylated, 1-phosphorylated, and protected with two S-acetyl-2-thioethyl groups[78] (Fig. 5a). Ac$_3$GlcNDAz-1P(Ac-SATE)$_2$ is cell-permeable and is metabolized to the nucleotide-sugar

UDP-GlcNDAz, which is accepted by OGT, resulting in the installation of O-GlcNDAz moieties onto native substrates[78]. Brief UV treatment of GlcNDAz-labeled cells affords the covalent, carbene-mediated, in situ cross-linking of O-GlcNDAz glycans to direct binding partner proteins within ~2–4 Å of the sugar (Fig. 5a)[78]. Therefore, Ac$_3$GlcNDAz-1P(Ac-SATE)$_2$ allows the capture, purification, and characterization of physiological O-GlcNAc-mediated PPIs[78].

We first performed Ac$_3$GlcNDAz-1P(Ac-SATE)$_2$ cross-linking on cells expressing WT NF-L. NF-L (~62 kDa predicted molecular weight; ~75 kDa on SDS-PAGE) cross-linked into ~200–250 kDa complexes in a diazirine- and UV-dependent manner (Fig. 5b). As expected[68], we also observed cross-linking of the heavily O-GlcNAcylated nucleoporin p62 (positive control) but not tubulin, which is not an OGT substrate (negative control), alongside low but detectable diazirine-independent background cross-linking from UV alone (Supplementary Fig. 4A). Co-expression of OGT potentiated Ac$_3$GlcNDAz-1P(Ac-SATE)$_2$ cross-linking of NF-L, whereas OGA co-expression reduced it (Fig. 5c), further supporting that the cross-links are specific and are mediated by O-GlcNAcylation. Notably, several steps in the Ac$_3$GlcNDAz-1P(Ac-SATE)$_2$ cross-linking procedure are less than 100% efficient, such as conversion to UDP-GlcNDAz, competition with abundant, endogenous UDP-GlcNAc, UV activation of the diazirine, and productive cross-linking prior to quenching of the carbene intermediate[78]. Therefore, our Ac$_3$GlcNDAz-1P(Ac-SATE)$_2$ cross-linking results almost certainly underestimate the prevalence or stoichiometry of O-GlcNAc-mediated PPIs of NF-L. These results indicate that NF-L engages in direct, O-GlcNAc-mediated PPIs.

We next identified the O-GlcNAc-mediated binding partners of NF-L. Because NF-L engages in both homotypic and heterotypic PPIs, we tested both types of interaction. First, we co-expressed two NF-L constructs with distinct myc and V5 epitope tags, performed Ac$_3$GlcNDAz-1P(Ac-SATE)$_2$ cross-linking, and analyzed the samples by myc IP and both myc and V5 IBs (Fig. 5d). NF-L molecules with each tag were present in the same cross-linked complexes (Fig. 5d), indicating that NF-L engages in homotypic O-GlcNAc-mediated PPIs. Second, to identify other O-GlcNAc-mediated binding partners of NF-L, we affinity-purified cross-linked complexes from vehicle- and Ac$_3$GlcNDAz-1P(Ac-SATE)$_2$-treated samples and analyzed them by MS. Interestingly, we observed INA in the +Ac$_3$GlcNDAz-1P(Ac-SATE)$_2$ sample (186 INA peptides by spectral count) but not in the vehicle control (0 INA peptides) (Supplementary Fig. 4B). INA is a

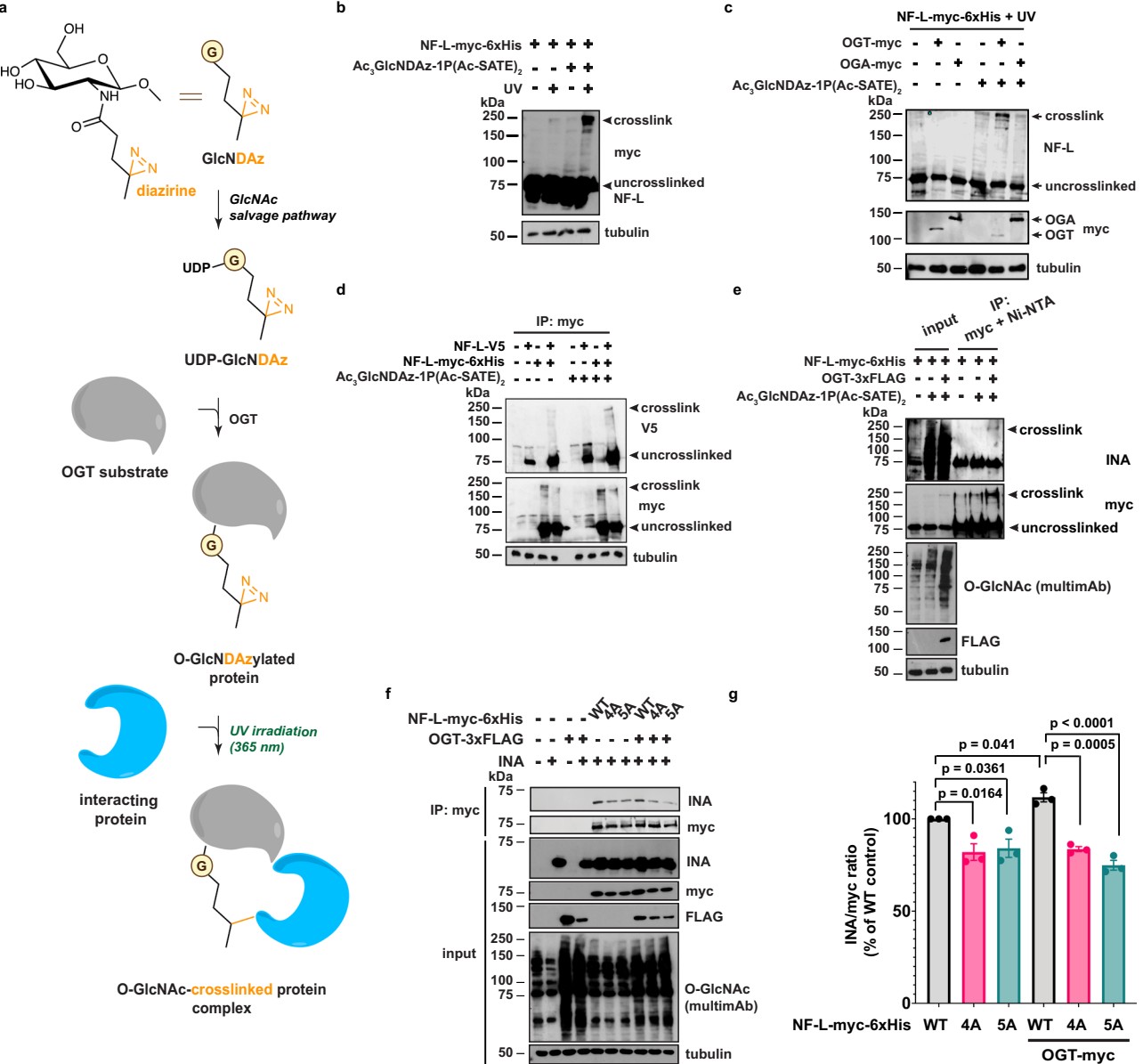

**Fig. 5 | Direct, O-GlcNAc-mediated interactions between NF-L and INA.**
**a** GlcNDAz, a GlcNAc analog that bears a diazirine photocross-linking moiety, can label OGT substrates in living cells. Brief UV treatment affords the covalent in situ cross-linking of O-GlcNDAz glycans to binding partner proteins, as described[78]. **b** 293T cells were transfected with NF-L-myc-6xHis ± 100 μM Ac₃GlcNDAz-1P(Ac-SATE)₂ for 48 h, subjected to UV cross-linking, and analyzed by IB (n = 3 biological replicates). **c** 293T cells were transfected with NF-L-myc-6xHis ± OGT-myc or OGA-myc ± 100 μM Ac₃GlcNDAz-1P(Ac-SATE)₂ for 48 h, subjected to UV cross-linking, and analyzed by IB (n = 2 biological replicates). **d** NEFL⁻/⁻ 293T cells were transfected with NF-L-myc-6xHis + NF-L-V5 ± 100 μM Ac₃GlcNDAz-1P(Ac-SATE)₂ for 48 h

and analyzed by myc IP and V5 IB (n = 3 biological replicates). **e** NEFL⁻/⁻ 293T cells were transfected with NF-L-myc-6xHis ± OGT-3xFLAG ± 100 μM Ac₃GlcNDAz-1P(Ac-SATE)₂ for 48 h and analyzed by tandem myc IP/Ni-NTA purification and IB with INA (EnCor Biotechnology) (n = 3 biological replicates). **f** NEFL⁻/⁻ 293T cells were transfected with NF-L-myc-6xHis, NF-L⁴ᴬ-myc-6xHis, or NF-L⁵ᴬ-myc-6xHis ± INA ± OGT-3xFLAG for 48 h and analyzed by myc IP and IB with INA (Novus Biologicals) (n = 3 biological replicates). **g** INA/myc ratio was calculated for the experiment described in (**f**). Data are shown as mean ± SEM and assessed by one-way ANOVA/Tukey's post hoc correction (n = 3 biological replicates).

comparatively low-abundance NF protein that heteropolymerizes with the triplet proteins[2] and aggregates in NF inclusion body disease, a form of frontotemporal dementia[79]. Tandem affinity purification experiments revealed that OGT expression increased the levels of both NF-L and INA in Ac₃GlcNDAz-1P(Ac-SATE)₂-cross-linked complexes (Fig. 5e), corroborating the MS results. Importantly, co-IP of WT NF-L and INA was potentiated by co-expression of OGT, confirming the O-GlcNAc-mediated interaction between these proteins in an Ac₃GlcNDAz-1P(Ac-SATE)₂-independent assay (Fig. 5f, g). Moreover, NF-L⁴ᴬ and NF-L⁵ᴬ exhibited reduced interaction with INA, compared to WT, and OGT co-expression failed to potentiate the interaction

between INA and either NF-L mutant (Fig. 5f, g). Together, these results demonstrate that O-GlcNAc moieties on NF-L engage in direct PPIs with other NF-L molecules and with INA, revealing glycan-mediated interactions within the NF network.

## NF-L O-GlcNAcylation is dysregulated by CMT-causative mutations

CMT-causative *NEFL* mutations trigger NF-L accumulation and aggregation in neurons[17–21], and WT NF protein aggregation is a feature of many other neurodegenerative disorders[16,22–25]. Given our results demonstrating that O-GlcNAc influences NF-L assembly state and the

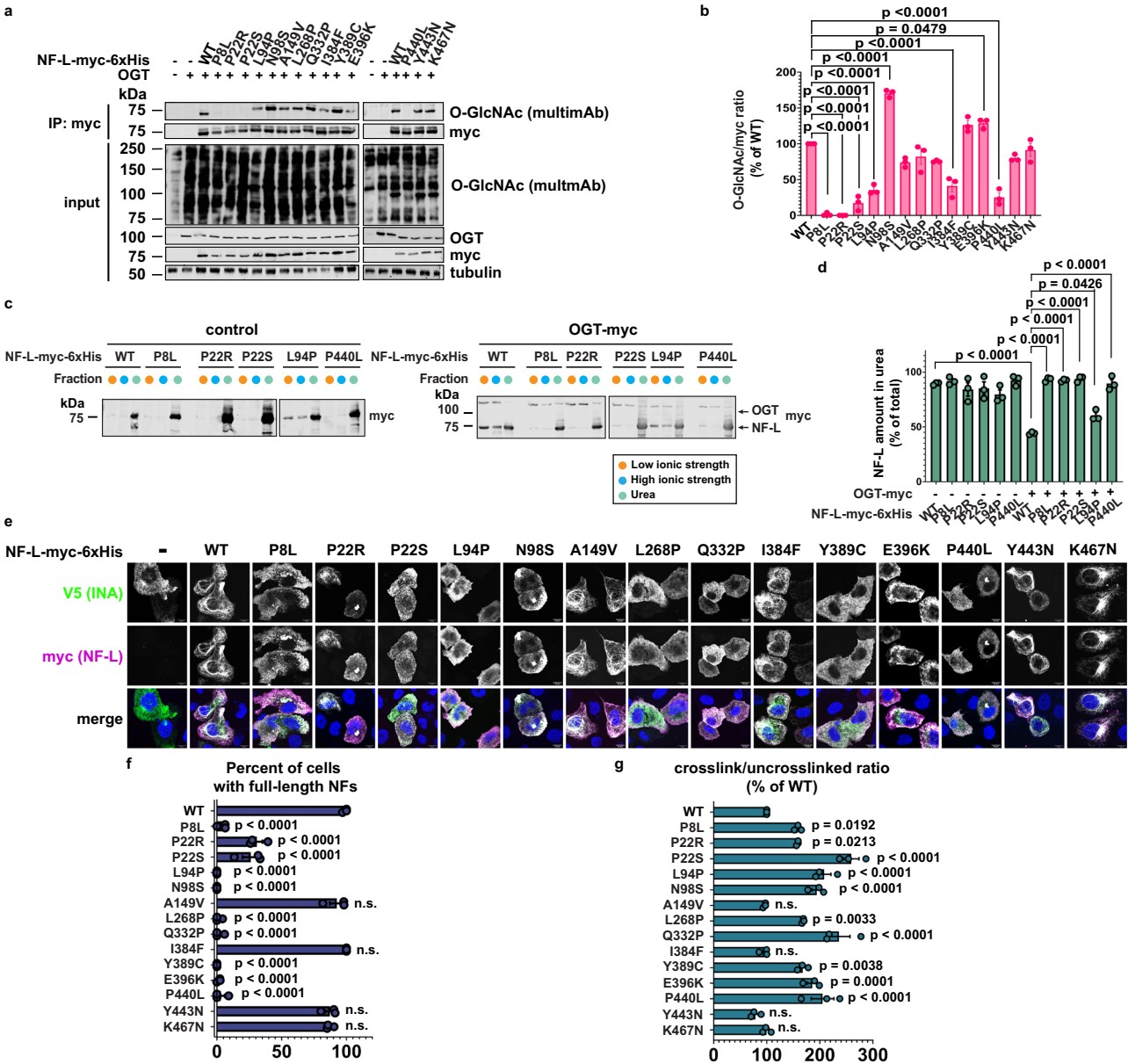

**Fig. 6 | NF-L O-GlcNAcylation is dysregulated by CMT-causative mutations.**
**a** *NEFL*[-/-] 293T cells were transfected with WT or CMT mutant NF-L-myc-6xHis + OGT for 24 h and analyzed by myc IP and IB. **b** Normalized O-GlcNAc signal (O-GlcNAc/myc ratio) was calculated for the experiments performed in (**a**). Data are shown as mean ± SEM and assessed by one-way ANOVA/Tukey's post hoc correction (*n* = 3 biological replicates). The samples from different gels derived from the same experiment, with sample processing controls (NF-L WT) included and gels/blots were processed in parallel. **c** *NEFL*[-/-] 293T cells were transfected with WT or CMT mutant NF-L-myc-6xHis ± OGT-myc for 24 h and analyzed by differential extraction and IB. **d** NF-L amount extracted into urea buffer was calculated as percent of total NF-L across three fractions from the experiment described in (**c**). Data are shown as mean ± SEM and assessed by one-way ANOVA/Tukey's post hoc correction (*n* = 3 biological replicates). The samples from different gels derived

from the same experiment, with sample processing controls (NF-L WT) included and gels/blots were processed in parallel. **e** SW13 vim[-] cells were transfected with WT or CMT mutant NF-L-myc-6xHis + INA-V5 for 24 h and analyzed by IFA. Scale bar: 10 μm. **f** Quantification of percent of cells with full-length NFs from the experiment described in (**e**) was performed by a blinded researcher. Data are shown as mean ± SEM and assessed by one-way ANOVA/Tukey's post hoc correction (*n* = 3 biological replicates). n.s. not significant. **g** *NEFL*[-/-] 293T cells were transfected with WT or CMT mutant NF-L-myc-6xHis ± 100 μM Ac₃GlcNDAz-1P(Ac-SATE)₂ for 48 h, subjected to UV cross-linking, and analyzed by IB. Normalized cross-link signal (cross-link/uncross-linked ratio) was calculated. Data are shown as mean ± SEM and assessed by one-way ANOVA/Tukey's post hoc correction (*n* = 3 biological replicates). n.s. not significant.

many well-established connections among O-GlcNAcylation, protein aggregation, and neurodegeneration in general[43], we tested whether NF dysfunction impacted NF-L O-GlcNAcylation. We generated NF-L constructs for 14 point-mutants that cause CMT[13] and quantified their O-GlcNAcylation by IP/IB (Fig. 6a, b). Most (8/14) mutants displayed significantly altered O-GlcNAcylation, compared to WT, with six hypoglycosylated and two hyperglycosylated (Fig. 6a, b). In particular, the four CMT mutations lying near NF-L glycosites (P8L, P22R, P22S,

P440L) greatly reduced or completely abolished NF-L O-GlcNAcylation (Fig. 6a, b). Because our results showed that NF-L O-GlcNAcylation modulates NF assembly state (Fig. 2) and mediates PPIs (Fig. 5), we hypothesized that hypoglycosylated CMT mutants would exhibit aberrant assembly. Consistent with literature reports demonstrating the formation of insoluble aggregates by CMT-causative NF-L mutants[21], differential extraction experiments revealed that all four hypoglycosylated NF-L CMT mutants were predominantly insoluble

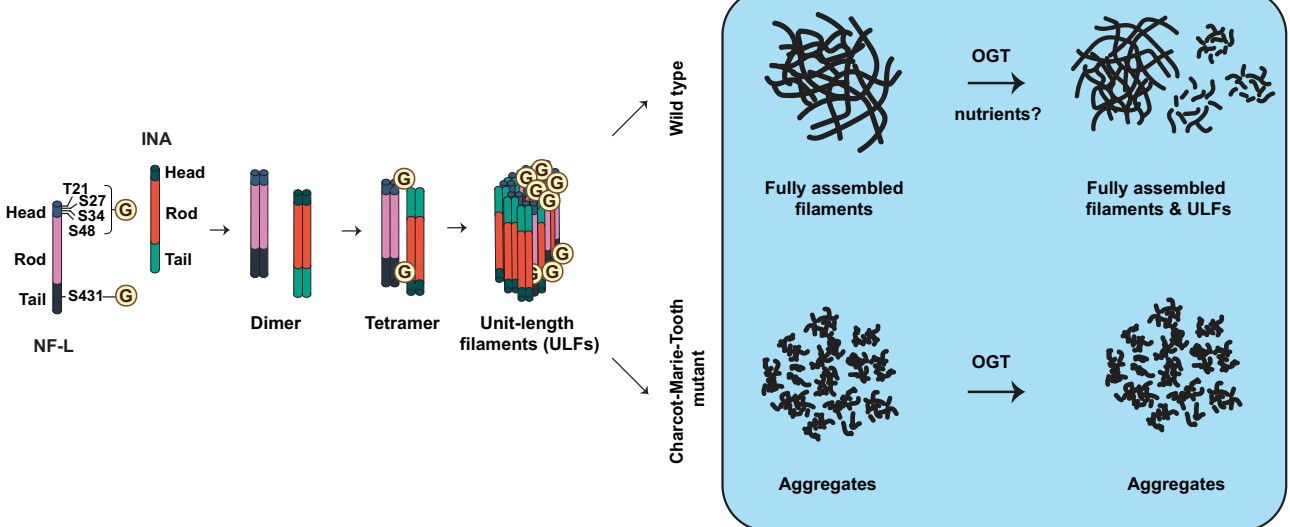

**Fig. 7 | Site-specific O-GlcNAcylation regulates NF-L assembly and function and is perturbed by CMT disease mutations.** Based on our results, we propose a model wherein nutrient-responsive O-GlcNAcylation of the NF-L head domain (T21/S27/S34/S48) promotes homotypic NF-L/NF-L and heterotypic NF-L/INA interactions and assembly and/or maintenance of assembled filaments under homeostatic conditions. Elevating NF-L O-GlcNAcylation reduces the prevalence of full-length filaments of WT NF-L, whereas CMT-causative mutations proximal to NF-L glyco-sites display lower O-GlcNAcylation and remain insensitive to the influence of OGT. The stoichiometry and topology of O-GlcNAc-mediated NF-L/INA interactions remain unknown. One plausible model is depicted for simplicity.

under basal conditions (Fig. 6c, d). Notably, elevating O-GlcNAc levels by OGT co-expression failed to drive these NF-L mutants into soluble assembly states, as it does for WT (Fig. 6c, d). To confirm and extend these findings, we performed IFA experiments in SW13 vim⁻ cells. In line with prior reports[15,72] and our own results, WT NF-L formed full-length, heteropolymeric filaments when co-expressed with INA, whereas most CMT mutants did not, instead forming punctate aggregates (Fig. 6e, f). Ac₃GlcNDAz-1P(Ac-SATE)₂ experiments showed aberrant, enhanced cross-linking by aggregate-forming NF-L CMT mutants, compared to WT or to the filament-forming mutants (A149V, I384F, Y443N, K467N) (Fig. 6g). All together, these data demonstrate that most CMT-causative mutants exhibit abnormal NF-L O-GlcNAcylation, PPIs, and assembly states. Based on our results, we propose a working model for the function of NF-L O-GlcNAcylation (Fig. 7), discussed below.

## Discussion

Nervous system function depends on the unique sizes, morphologies, and intricate subcellular organization of individual neurons[1,3–6]. These properties, in turn, rely on a complex and dynamic neuronal cytoskeleton[1,3–6]. In particular, the NF network is required for neuronal shape, mechanical integrity, organelle trafficking, and synapse architecture[2,26]. In addition to their physiological importance, the dysregulation of NF proteins contributes to myriad neurological diseases[13,16,22–25]. Therefore, understanding the dynamic regulation of the NF cytoskeleton is a crucial—but incompletely realized—goal.

Here we show that NF-L, which is essential for NF formation[2,26], is regulated by site-specific O-GlcNAcylation in cultured cells, primary neurons, and human brain tissue. Our MS site-mapping and muta-genesis data pinpointed five O-GlcNAc sites on human NF-L, predominantly in the head domain (Fig. 1f). These results agree with prior proteomics studies of rodent NF-L orthologs[31–33] and post-mortem human brain samples[34], and we have extended this earlier work by validating and characterizing the function of human NF-L O-GlcNAcylation.

The regulation of human NF-L by O-GlcNAc is likely even more extensive than our current results indicate. For example, consistent with prior work[34], our MS data detected an additional high-confidence O-GlcNAcylated peptide in the NF-L tail domain (residues 404–416), but the specific glycosite was not identifiable, due to low abundance. (Complete MS O-GlcNAc site-mapping data are provided—please see "Methods" and "Data availability".) Therefore, NF-L O-GlcNAcylation probably occurs on more residues than we have characterized thus far, perhaps explaining the residual O-GlcNAc signal detected on the NF-L[5A] mutant (Fig. 1h, j). Our results on human NF-L may also be relevant to other NF proteins and to other species. For example, some NF-L gly-cosites characterized here are conserved among vertebrate NF-L orthologs, and cognate residues in human NF-M (e.g., T19 and T48, corresponding to NF-L T21 and S48, respectively) are reportedly O-GlcNAcylated in mouse neurons[31,32]. Beyond NFs, we[68] and others[80,81] have previously demonstrated the functional importance of O-GlcNAcylation for the assembly states and functions of other IF proteins, implying that O-GlcNAc may be a general regulator of the > 70-member human IF protein family. Hence, thorough characterization of O-GlcNAcylation on NF-L and other NF proteins may provide insight into IF protein regulation in other tissues.

Our results also align with prior studies demonstrating the regulation of NF proteins by PTMs in general[30,66,82]. For example, NF proteins are among the most highly phosphorylated substrates in the brain[33]. Phosphorylation of the NF-L head domain regulates NF assembly[66] and NF transport towards dendrites[82], and phosphorylation of the tail domains of NF-M and NF-H impacts the regular spacing between assembled NFs[67]. Aberrant NF phosphorylation is a patho-logical feature of several human neurodegenerative diseases, including CMT type 2E[74], AD[83], and ALS[84]. Our results show that O-GlcNAcylation, like phosphorylation, can influence NF-L assembly state and down-stream functions, such as the regulation of organelle transport (Fig. 3).

Interestingly, there is complex and well-documented crosstalk between phosphorylation and O-GlcNAcylation in many contexts[33,62,63]. For example, the two PTMs can compete for identical or nearby resi-dues as modification sites or, conversely, the modification of one residue by one PTM can promote the modification of a nearby residue by the other[62,63]. In addition, OGT, OGA, kinases, and phosphatases modify each other and are often detected together in multiprotein complexes, constituting another layer of crosstalk[62,63]. Our data sug-gest that O-GlcNAc/O-phosphate interplay may regulate NF-L, as these

PTMs are reciprocal under the conditions tested thus far (Supplemental Fig. 1F, G). However, these phosphorylations likely occur at distinct, as-yet unknown residues, because none of the NF-L glycosites that we identified is a documented phosphosite[33]. The precise biochemical basis and regulatory implications of these observations will be an important goal of subsequent mechanistic and phenotypic studies.

Functionally, we demonstrate that NF-L O-GlcNAcylation is required for the regulation of mitochondrial and lysosomal motility (Fig. 3). Our results are consistent with prior reports of NFs influencing mitochondrial trafficking[10,15,18,20]. For example, in human iPSC-derived motor neurons, mitochondrial movement is increased in the absence of NF-L[10] and decreased in the presence of NF-L mutants that aggregate in the soma[20]. In our experiments, the expression of WT NF-L suppressed mitochondrial motility, but expression of the NF-L[4A] glycosite mutant, which exhibits assembly state defects (Fig. 2e, f and Supplementary Fig. 2K, L), failed to do so (Fig. 3a, b). The endogenous microtubule cytoskeleton is intact under these conditions (Supplementary Fig. 3A), indicating that gross microtubule disruption cannot explain the NF-L mutant phenotype. Instead, we propose that WT NF-L incorporates into the endogenous NFs, creating a robust network in the axonal cytoplasm that mitochondria must circumnavigate, whereas loss of NF-L O-GlcNAcylation dysregulates NF assembly state, reducing these barriers and accelerating mitochondria motility. Perhaps counter-intuitively, WT NF-L expression accelerated lysosomal motility in our system, whereas NF-L[4A] did not (Fig. 3c, d). It may be that comparatively small lysosomes benefit from less crowding by impeded mitochondria in the presence of a more elaborate WT NF network, but this hypothesis remains to be tested. Though one prior report found no effect of NF-L loss on lysosomal motility[10], differences in experimental systems may account for this ostensible discrepancy with our work.

Mitochondria and lysosomes are transported along microtubules in the forward direction by kinesins and in the reverse by dyneins[77]. In principle, NF-L glycosylation could affect the action of one or both motor complexes. Further work will be required to determine whether kinesin- and dynein-dependent motility are differentially affected by NF O-GlcNAcylation in other contexts or model systems. Regardless, our data consistently indicate that NF-L O-GlcNAcylation impacts organelle motility in primary neurons.

Mechanistically, earlier studies proposed a role for O-GlcNAcylation in regulating rodent NF assembly[31,32]. Our results support this hypothesis by demonstrating that O-GlcNAc impacts human NF-L function at least in part by modulating its assembly state and PPIs. Biochemical and imaging experiments showed that O-GlcNAcylation drives WT NF-L to lower-order assembly states, reducing the prevalence of full-length NFs (Fig. 2). Furthermore, our Ac$_3$GlcNDAz-1P(Ac-SATE)$_2$ cross-linking experiments showed that NF-L engages in both homotypic (NF-L/NF-L) and heterotypic (NF-L/INA) O-GlcNAc-mediated interactions (Fig. 5), providing a potential mechanism to explain the effects of NF-L O-GlcNAcylation on NF assembly state (Fig. 7). These results revealed previously unknown, glycan-mediated biochemical interactions among NF proteins and provide new information on the molecular regulation of the NF cytoskeleton.

Despite these insights, additional questions remain. For example, the stoichiometry and structural basis of O-GlcNAc-mediated NF-L PPIs are as yet unknown. Based on the apparent molecular weight of GlcNDAz-dependent cross-links (~250 kDa) and the evidence for homotypic NF-L/NF-L PPIs (Fig. 5d), we propose that these complexes contain four or five distinct polypeptides, including at least two NF-L (62 kDa) and at least one INA (55 kDa) molecules. This model is broadly consistent with the established 4:1 ratio of NF-L to INA in mature NFs[85]. Because increased O-GlcNAcylation potentiates NF-L/INA interactions (Fig. 5e–g) and reduces the prevalence of fully assembled NFs (Fig. 2b–f), we hypothesize that O-GlcNAc-mediated homotypic and

heterotypic PPIs of NF-L occur within lower-order assembly states in vivo. We demonstrated that ablating all five identified human NF-L glycosites significantly decreased NF-L/INA interactions (Fig. 5f, g), indicating that O-GlcNAcylation of NF-L itself is at least partly responsible for these interactions. Whether INA is also O-GlcNAc-modified has not been specifically examined, but untargeted glycoproteomics studies have reported data suggesting that it might be[33,34,40]. O-GlcNAc moieties on INA could hypothetically contribute to interactions with NF-L. Because INA exhibits aberrant PPIs in NF inclusion body disease[79], these questions may also have clinical relevance, an interesting hypothesis to test in later work. However, the paucity of information on INA O-GlcNAcylation and the total lack of structural information on NF proteins[86] currently hampers a more detailed characterization of these interactions. Additional biophysical, MS, and in vivo experiments in future studies will be needed to define the stoichiometry, structural underpinnings, and pathophysiological significance of O-GlcNAc-mediated PPIs among NF components, such as INA.

Our results indicate that upstream signals could regulate NF-L by inducing or inhibiting its O-GlcNAcylation (Figs. 1–3 and Supplementary Figs. 1 and 2), and we identify nutrient or growth factor availability as candidate stimuli that may govern NF-L O-GlcNAcylation in vivo (Fig. 4). Although further studies will be needed to dissect the relationship between nutrient-sensing and NF-L glycosylation in neurons, similar phenomena have been reported in other systems. For example, the Hart lab previously showed that glucose deprivation alters the O-GlcNAcylation of NF-H in a p38 mitogen-activated protein kinase-dependent manner, leading to increased NF-H solubility[87]. In other contexts, the Schwarz lab demonstrated that local glucose concentration differences within the axon regulate mitochondrial motility through the O-GlcNAcylation of microtubule-dependent motor complex components[88]. Experiments are currently underway to determine the impact of nutrient and growth factor changes on NF-L O-GlcNAcylation and function, including organelle motility, in cultured primary neurons or human iPSC-derived neurons.

Finally, our work examines the potential relationships among NF-L O-GlcNAcylation, mis-assembly, and CMT mutations[13]. A longstanding puzzle in the CMT field is that mutations throughout the NF-L protein sequence can cause disease, yet no clear correlations have emerged among *NEFL* genotypes and molecular, cellular, or clinical phenotypes, despite extensive characterization of a large number of mutations[13]. Prior studies showed that some CMT-causative *NEFL* mutations severely impact NF axonal transport[19], subcellular mitochondrial distribution, or neuronal processes in cultured cells[15–21]. A prevalent hypothesis is that the assembly defects of NF-L mutants observed in cultured cells[19], mouse brains[17], and human iPSC-derived motor neurons[18] lead to NF aggregation and consequent axonal dysfunction[13]. However, the phenotypic outcomes of NF mis-assembly vary significantly among CMT *NEFL* mutations, with no obvious, unifying, coherent pattern[13].

In some respects, our results reflect a similar conundrum. For example, most CMT NF-L mutants we tested exhibited altered O-GlcNAcylation, compared to WT (Fig. 6a, b). However, decreased, increased, and unchanged glycosylation phenotypes were all observed (Fig. 6a, b) and did not consistently correlate with assembly state defects (Fig. 6c–f). Similarly, most of the CMT mutants we tested abrogated NF formation, whereas others (A149V, I384F, Y443N, K467N) were still able to form filaments with INA (Fig. 6e–g). These ostensibly divergent observations highlight the well-known challenge and importance of continuing to study CMT *NEFL* mutations to understand their molecular defects[13].

On the other hand, we observed two intriguing correlations between NF-L O-GlcNAcylation and downstream phenotypes among CMT mutants. First, all NF-L CMT variants with mutations near O-GlcNAc sites were hypoglycosylated and resisted the OGT-induced

shift to lower-order assembly states observed with WT NF-L (Fig. 6c, d). These mutations may alter the primary sequence or secondary structural determinants required by OGT to bind NF-L and/or may adopt abnormal tertiary or quaternary structures that occlude OGT's access to its target residues. Consistent with this notion, a recent study showed that the NF-L head domain, which is rich in low-complexity sequence[89], self-associates via labile cross-β structures[89]. In this work, residues P8 and P22, which are CMT mutational hotspots near glycosites[13], reduced the formation of cross-β structures[89], and mutations at these positions resulted in enhanced polymerization and head-domain self-association[89]. Second, most CMT mutants that aggregated also demonstrated enhanced levels of Ac$_3$GlcNDAz-1P(Ac-SATE)$_2$ cross-linking, relative to WT (Fig. 6g). In contrast, the CMT mutants that could still form filaments (A149V, I384F, Y443N, K467N) exhibited near-WT levels of Ac$_3$GlcNDAz-1P(Ac-SATE)$_2$ cross-linking (cf. Fig. 6e–g). These findings suggest that particularly aggregation-prone CMT mutants may adopt irregular conformations[13] that predispose them to increased non-specific cross-linking in the Ac$_3$GlcNDAz-1P(Ac-SATE)$_2$ assay. It will be important to determine in future work whether loss of O-GlcNAcylation on these CMT mutants promotes their adoption of aberrant conformations in vivo, potentially leading to aggregation and downstream pathological effects.

Overall, our results consistently indicate that most CMT mutations cause anomalies in NF-L O-GlcNAcylation, assembly state, and PPIs. Moreover, our study suggests a potential future diagnostic benefit of characterizing NF-L O-GlcNAcylation or particular glycoforms in diseases characterized by NF dysregulation, such as AD[22], PD[23], ALS[24], GAN[16], and spinal muscular atrophy[25], in addition to CMT. Sensitive assays of NF-L in CSF and blood have shown great promise as biomarkers for a range of nervous system conditions[2,26]. In this context, our results may point to future opportunities to exploit particular glycoforms of NF-L in disease diagnostics.

## Methods

### Ethical approval

All animal work in this study was performed under the oversight of the Duke University Institutional Animal Care and Use Committee, which reviewed and approved the written protocol (#A230-21-11) and regularly inspects all animal facilities. Rats were acquired through Duke's Division of Laboratory Animal Resources, which has expert veterinary staff to attend to their daily care and well-being. Rats were housed in a Duke animal care facility accredited by the Association for Assessment and Accreditation of Laboratory Animal Care (AAALAC). All animal procedures, including euthanasia, were carried out according to the general guidelines of the US Animal Welfare Act and AAALAC and complied fully with all relevant ethical standards.

### Chemicals

Thiamet-G was purchased from Cayman Chemical (#13237). Peracetylated 5SGlcNAc and Ac$_3$GlcNDAz-1P(Ac-SATE)$_2$ were synthesized by the Duke Small Molecule Synthesis Facility essentially as described[56,78].

### DNA constructs

The mito-mEmerald and LAMP1-GFP constructs have been described[75]. NF-L-V5/plenti6.3-DEST was purchased (DNASU, HsCD00870013). Mutagenic primers were designed with QuikChange Primer Design (Agilent). To make NF-L-myc-6xHis glycosite mutants, NF-L-mCherry, NF-M-GFP, NF-H-3xFLAG, OGT-3xFLAG, INA-V5, and UAP1$^{F383G}$-FLAG constructs, PCR was performed using Phusion Hot Start II DNA polymerase (ThermoFisher Scientific, F549S) with primers included in Supplementary Table 1. PCRs were digested with DpnI (New England Biolabs [NEB], R0176S), purified using a gel DNA recovery kit (Zymo Research, D4002), and ligated using NEBuilder HiFi DNA Assembly Cloning Kit (NEB, E5520S) with 1:3 vector:insert mass ratio calculated by NEBioCalculator (https://nebiocalculator.neb.com/#!/ligation). Then, the ligated product was transformed in 10-beta competent *E. coli* (NEB, C3019H), and colonies were picked for maxipreps (ZymoPURE II Plasmid Purification Kit, Zymo Research, D4202) and Sanger sequencing. Guide RNAs (gRNAs) for *NEFL* knockout (5'-CTCGTAGCTGAAG-GAACTCA-3') and *NEFL* knock-in (5'-GTAGCTGAAGGAACTCATGG-3') were designed by DESKGEN tool and subcloned into pSpCas9(BB)-2A-GFP (Addgene, 48138) with BbsI-HF (NEB, R3539) and T4 DNA ligase (NEB, M0202S). For the *NEFL* knock-in repair template, the 3xFLAG-6xHis gene fragment (IDT) was subcloned into the *AAVS1*_3xFLAG-2xStrep plasmid (Addgene, 68375) at the NcoI/BstBI sites ("*AAVS1*_3x-FLAG-6xHis"). The *NEFL* homology arm (HA)_1 fragment (Genscript) was cloned into the *AAVS1*_3xFLAG-6xHis vector at the Nde1/NcoI sites ("*AAVS1*_3xFLAG-6xHis-HA_1"). The HA_2 fragment (Genscript) was then ligated to the *AAVS1*_3xFLAG-6xHis-HA_1 vector at the BstBI/EcoRI sites. The nucleotide sequences for the NF-L$^{4A}$ gene fragment (IDT), 3xFLAG-6xHis, and two HA fragments are included in the Supplementary Table 2.

### Cell culture

293T (ATCC CRL-11268) and SW13 vim$^-$ cells (Snider lab, UNC) were grown in Dulbecco's modified Eagle Medium (DMEM, Sigma-Aldrich, D6429) containing 10% fetal bovine serum (FBS, Sigma-Aldrich, F0926), 100 units/mL penicillin, and 100 μg/mL streptomycin (Pen/Strep, Gibco, 15140-122). SH-SY5Y (ATCC CRL-2266) cells were grown in DMEM/F12 (Gibco, 11330-032) containing 10% FBS and 1% Pen/Strep. After CRISPR manipulations, single GFP-positive cells were sorted by flow cytometry and plated into DMEM (293T) or DMEM/F12 with 15% FBS, 1% Pen/Strep (SH-SY5Y). For sorted SH-SY5Y cells, DMEM/F12 was changed every two days. For nutrient starvation, *NEFL*$^{-/-}$ 293T cells expressing NF-L-myc-6xHis were starved for 48 h of glucose: glucose-free DMEM (ThermoFisher Scientific, 11966025) with 1 mM sodium pyruvate (ThermoFisher Scientific, 11360070), 10% FBS, and Pen/Strep; glutamine: glutamine-free DMEM (ThermoFisher Scientific, 11960044) with 1 mM sodium pyruvate, 10% FBS, and Pen/Strep; or serum: DMEM with 1 mM pyruvate and Pen/Strep. For hippocampal neurons, embryonic day 18 Sprague Dawley rat hippocampi were isolated as described[75]. Neurons were plated on 35 mm culture dishes precoated with 0.5 mg/mL poly-L-lysine (Sigma-Aldrich, P1274) at a density of 125,000 cells/dish. Neurons were initially plated in minimum essential medium (Gibco, 11095-072) supplemented with 10% horse serum (Gibco, 16050122), 33 mM D-glucose (Sigma, G8769-100), and 1 mM sodium pyruvate (Corning, 25-000Cl) and incubated for 2–5 h. The medium was then replaced with Neurobasal (Thermo Scientific, 21103049) supplemented with 33 mM D-glucose, 2 mM GlutaMAX (Life Technologies, 35050061), 1% Pen/Step, and 2% B-27 (ThermoFisher, 12-587-010). Cytosine arabinoside (5 μM) was added the day after plating to prevent glial cell proliferation. All cell lines were maintained at 37 °C in a 5% CO$_2$ atmosphere.

### Transfections

Unless otherwise indicated, cells were transfected with 10 μg of DNA at 70–80% cell density using TransIT-LT1 (Mirus, 2300) for 293T cells and at 50–60% cell density with Lipofectamine 3000 (ThermoFisher Scientific, L3000001) for SH-SY5Y cells. For co-IPs, 293T cells were transfected with 8 μg of each DNA at 25–30% cell density. For CRISPR-tagging of *NEFL*, 10 million 293T cells seeded on a 15-cm culture dish were transfected with 10 μg of pSpCas9-GFP-*NEFL* knock-in (or pSpCas9-GFP-*AAVS1*, negative control) single gRNA (gRNA) construct and 20 μg of *NEFL* homology-directed repair vector at 50–60% cell density. For CRISPR deletion, 10 million 293T cells in a 10-cm culture dish or 10 million SH-SY5Y cells in a 15-cm culture dish were transfected with 15 μg of pSpCas9-GFP-*NEFL* knockout (or pSpCas9-GFP-*AAVS1*, negative control) gRNA construct.

## IP/co-IP

Twenty-four hours (or 48 h for co-IPs) post-transfection, cells were harvested in cold phosphate-buffered saline (PBS) and lysed in cold IP lysis buffer (20 mM Tris-HCl pH 7.4, 1% Triton X-100, 0.1% SDS, 150 mM NaCl, 1 mM EDTA) with protease inhibitor cocktail (Sigma, P8340, 1:100), 200 µM $Na_3VO_4$ (Millipore Sigma, 13721-39-6), 50 µM UDP (Sigma, 94330; OGT inhibitor), and 5 µM PUGNAc (Cayman Chemical, 17151; OGA inhibitor). Lysates were incubated on ice for 15 min, probe-sonicated for 50 s at 40% duty cycle, and cleared by centrifugation at $27,000 \times g$ for 15 min at 4 °C. Cleared lysates were quantified by bicinchoninic acid (BCA) assay (ThermoFisher, 23225). IPs/co-IPs were performed on 1.5–2 mg of total protein in 0.5 mL (~3-4 mg/mL protein). Unless otherwise indicated, 3 µg of primary antibody per 1 mg of protein was added to the protein lysate for rotation overnight at 4 °C. The next day, 20 µL of settled protein A/G UltraLink resin (Thermo-Fisher, 53133) were washed three times in the corresponding lysis buffer and added to each IP/co-IP for rotation at 4 °C for 2 h. Proteins were eluted in lysis buffer supplemented with 5% fresh β-mercaptoethanol (Sigma, M3148) and 1× SDS-PAGE loading buffer (5× SDS-PAGE loading buffer: 250 mM Tris pH 6.8, 10% SDS, 30% glycerol, 5% β-mercaptoethanol, 0.02% bromophenol blue). Eluates were heated at 95 °C for 5 min and analyzed by IB.

## IB

For enhanced chemiluminescence (ECL) detection, SDS-PAGE gels were electroblotted onto 100% methanol pre-soaked polyvinylidene difluoride membranes (PVDF, 0.45 µm, ThermoFisher Scientific, 88518) in transfer buffer (25 mM Tris pH 8, 192 mM glycine, 0.1% SDS, 20% methanol) using a BioRad TransBlot Turbo system. Then, membranes were incubated in blocking buffer (2.5% (w/v) bovine serum albumin [BSA] in Tris-buffered saline with Tween [TBST] [20 mM Tris-HCl pH 7.4, 150 mM NaCl, 0.1% Tween 20]) with agitation at room temperature (RT) for 30 min. Membranes were incubated with primary antibody diluted 1:1000 (except for α-tubulin, 1:100,000; INA, Novus Biologicals, 1:2000) in blocking buffer overnight with gentle shaking at 4 °C. The next day, membranes were washed three times with TBST, each 10 min, and incubated with the appropriate horseradish perox-idase (HRP)-conjugated secondary antibody diluted 1:10,000 in blocking buffer at RT for 1 hr. Membranes were again washed three times with TBST, each 10 min. Bands were visualized using enhanced chemiluminescence (Genesee Scientific 20-300B) and photographic film (LabScientific, XAR ALF 2025). For quantitative fluorescent IBs, gels were electroblotted onto a nitrocellulose membrane (0.45 µm, BioRad, 1620115). Blocking, primary, and washing conditions were the same as above. Membranes were incubated with appropriate IRDye-conjugated secondary antibody diluted 1:30,000 in blocking buffer at RT in the dark for 1 h. Membranes were washed three times with TBST, each 5 min. Bands were visualized using a LI-COR Odyssey DLx Imaging System. Complete antibody information is provided in Supplementary Table 3.

## NF-L O-GlcNAcylation

For endogenous NF-L O-GlcNAcylation, SH-SY5Y cells cultured in 50 µM Thiamet-G were lysed in 8 M urea/PBS on ice for 30 min, homogenized by probe-sonication for 50 s at 40% duty cycle, and exchanged to IP lysis buffer using Zeba spin desalting columns (ThermoFisher Scientific, 89892). For brain NF-L O-GlcNAcylation, post-mortem human brain chips (Duke Bryan Brain Bank and Bior-epository) were thawed on ice for 3–4 h, lysed in 2 mL of ice-cold 8 M urea/PBS by pipetting, and rotated for 20 min at 4 °C. Samples were probe-sonicated on ice with 40% duty cycle for 90 s, 50% for 90 s, and 60% for 30 s. Lysates were cleared by centrifugation at $27,000 \times g$, 30 min at 4 °C, and the supernatant lipid layer was removed. Cleared lysates were transferred to a new microfuge tube. The remaining pellet was again extracted in 0.5 mL of ice-cold 8 M urea/PBS as above. The

cleared samples were combined, exchanged into IP lysis buffer (20 mM Tris-HCl pH 7.4, 1% Triton X-100, 0.1% SDS, 150 mM NaCl, 1 mM EDTA) supplemented with protease inhibitor cocktail (1:100), 200 µM $Na_3VO_4$, 50 µM UDP, and 5 µM PUGNAc using a Zeba column, quanti-fied by BCA, and analyzed by IP/IB.

## On-blot β-elimination

Brain homogenates were run in duplicate on an SDS-PAGE gel and electroblotted onto a single PVDF membrane as above. The membrane was cut in half vertically, with one half incubated in 55 mM NaOH and the other in water (control) at 40 °C with rocking, as described[57]. Membranes were then washed three times in TBS, blocked with 2.5% (w/v) BSA in TBST at RT for 1 h, and analyzed by IB.

## CRISPR tagging or deletion of *NEFL*

Forty-eight hours post-transfection with the constructs listed above, single GFP-positive cells were sorted into 96-well plates using a DiVa fluorescence-activated cell sorter (BD Biosciences) at the Duke Cancer Institute Flow Cytometry Shared Resource. Sorted cells were expan-ded into single cell-derived clones and verified for NF-L-3xFLAG-6xHis expression by reciprocal IP/IB or for *NEFL* deletion by quantitative RT-PCR (qPCR), IP/IB, and IFA with NF-L antibody. Validated clones of NF-L-3xFLAG-6xHis-expressing 293T, *NEFL*$^{-/-}$ 293T, and *NEFL*$^{-/-}$ SH-SY5Y were selected for subsequent experiments.

## qPCR

Parental or *NEFL*$^{-/-}$ SH-SY5Y cells seeded in a six-well plate were col-lected at 80% cell density, lysed, and RNA-extracted using the RNeasy Mini Kit (Qiagen, 74104). RTL buffer was supplemented with β-mercaptoethanol prior to lysis. RNA concentration was measured by Nanodrop 2000 Spectrophotometer (ThermoFisher Scientific, ND-2000). *NEFL* cDNAs were made using SuperScript II reverse tran-scriptase II (ThermoFisher Scientific, 18064014) and quantified by qPCR. The qPCR primer pairs (Supplementary Table 1) were com-plementary to different regions of the *NEFL* mRNA. Relative levels of *NEFL* mRNA were normalized to β-actin mRNA in each sample. Tripli-cate reactions were performed using Sybr Master Mix (Life Tech-Power Sybr, 4367659) and a StepOnePlus Real-Time PCR system (Applied Biosystems).

## Endogenous NF-L purification for glycosite mapping

NF-L-3xFLAG-6xHis 293T cells were treated with 50 µM Thiamet-G and 4 mM glucosamine for 24 h as described[59,60]. Then, cells from 45 15-cm culture dishes were harvested, lysed, and quantified as for IP/co-IP. 270 mg total protein was divided evenly into 11 5-mL centrifuge tubes (Genesee Scientific, 24-285) and rotated with FLAG antibody overnight at 4 °C. In total, 75 µL settled, washed protein A/G UltraLink resin was then added and rotated at 4 °C for 2 h, then all tubes were pooled. Resin was washed with IP lysis buffer without EDTA (20 mM Tris-HCl pH 7.4, 1% Triton X-100, 0.1% SDS, 150 mM NaCl) five times and rotated in 600 µL buffer A (300 mM NaCl, 1% Triton X-100, and 10 mM imi-dazole in 8 M urea/PBS) twice, each for 30 min, at 4 °C. After cen-trifugation (2500×g, 5 min), the total cleared supernatant (1.2 mL) was rotated with 100 µL of HisPur Ni-NTA resin (ThermoFisher, 88223) for 4 h at 4 °C. Resin was then washed ten times with buffer A and eluted five times, each in 60 µL of elution buffer (250 mM imidazole in 8 M urea/PBS) for 20 min with vigorous shaking at RT. Purified NF-L sam-ples were processed either from colloidal blue-stained (ThermoFisher Scientific, LC6025) SDS-PAGE gel band or directly in-solution from the eluate.

## Liquid chromatography-tandem MS analysis

For gel band analysis, colloidal blue-stained SDS-PAGE bands (Invi-trogen Unpaged 4–12% Bis-Tris) were manually excised and subjected to reduction, alkylation, and in-gel tryptic digestion as described[59,60].

For in-solution analysis, samples were supplemented with 5% SDS, reduced with 10 mM DTT for 15 min at 80 °C, alkylated with 20 mM iodoacetamide for 30 min at RT, then supplemented with a final concentration of 1.2% phosphoric acid and 609 μL of S-Trap (Protifi) binding buffer (90% MeOH/100 mM triethylammonium bicarbonate [TEAB]). Proteins were trapped on an S-Trap micro cartridge, digested using 20 ng/μL sequencing-grade trypsin (Promega) for 1 h at 47 °C, and eluted using 50 mM TEAB, followed by 0.2% formic acid (FA), followed by 50% acetonitrile (ACN)/0.2% FA. All samples were then lyophilized to dryness. Dried samples were subjected to chromatographic separation on a Waters NanoAquity UPLC equipped with a 1.7 μm BEH130 C18 75 μm I.D. × 250 mm reversed-phase column. The mobile phase consisted of (A) 0.1% FA in water and (B) 0.1% FA in ACN. Following a 4 μL injection, peptides were trapped for 3 min on a 5 μm Symmetry C18 180 μm I.D. × 20 mm column at 5 μL/min in 99.9% A. The analytical column was then switched in-line, and a linear elution gradient of 5% B to 40% B was performed over 60 min at 400 nL/min. The analytical column was connected to a fused silica PicoTip emitter (New Objective) with a 10-μm tip orifice and coupled to a Lumos mass spectrometer (Thermo Scientific) through an electrospray interface operating in data-dependent acquisition mode. The instrument was set to acquire a precursor MS scan from $m/z$ 350 to 1800 every 3 s. In data-dependent mode, MS/MS scans of the most abundant precursors were collected at $r = 15,000$ (45 ms, AGC 5e4) following higher-energy collisional dissociation (HCD) fragmentation at an HCD collision energy of 27%. Within the MS/MS spectra, if any diagnostic O-GlcNAc fragment ions ($m/z$ 204.0867, 138.0545, or 366.1396) were observed, a second MS/MS spectrum at $r = 30,000$ (250 ms, 3e5) of the precursor was acquired with electron transfer dissociation (ETD)/HCD fragmentation using charge-dependent ETD reaction times and either 30% (2+ charge state) or 15% (3+ to 5+ charge state) supplemental collision energy.

For all experiments, a 60 s dynamic exclusion was employed for previously fragmented precursor ions. Raw liquid chromatography-tandem MS (LC-MS/MS) data files were processed in Proteome Discoverer (ThermoFisher Scientific) and then submitted to independent Mascot searches (Matrix Science) against a SwissProt database (human taxonomy) containing both forward and reverse entries of each protein (https://www.uniprot.org/proteomes/UP000005640) (20,322 forward entries). Search tolerances were 2 ppm for precursor ions and 0.02 Da for product ions using semi-trypsin specificity with up to two missed cleavages. Both y/b-type HCD and c/z-type ETD fragment ions were allowed for interpreting all spectra.

Carbamidomethylation (+57.0214 Da on C) was set as a fixed modification, whereas oxidation (+15.9949 Da on M), phosphorylation (+79.97 Da on S/T), and O-GlcNAc (+203.0794 Da on S/T) were considered dynamic mass modifications. All searched spectra were imported into Scaffold (v4.1, Proteome Software), and scoring thresholds were set to achieve a peptide false discovery rate of 1% using the PeptideProphet algorithm (http://peptideprophet.sourceforge.net/). When satisfactory ETD fragmentation was not obtained upon manual inspection, HCD fragmentation was used to determine O-GlcNAc residue modification using the number of HexNAcs identified in combination with the number of S/T residues in the peptide.

## Ac₃GlcNDAz-1P(Ac-SATE)₂ cross-linking

Ac₃GlcNDAz-1P(Ac-SATE)₂ experiments were performed essentially as described[78]. 293T cells were transfected with UAP1^F383G-myc-6xHis and NF-L-myc-6xHis for 24 h, then treated with DMSO or 100 μM Ac₃GlcNDAz-1P(Ac-SATE)₂ every 24 h over 48 h. Just before cross-linking, medium was removed and replaced with PBS. With lids removed, culture dishes were placed on ice and underneath a 365 nm UV light (Blak-Ray XX-20BLB UV Bench Lamp, 95-0045-04) for 25 min. Cells were scraped into cold PBS and lysed in 8 M urea/PBS, and lysates were analyzed by IB.

## Ac₃GlcNDAz-1P(Ac-SATE)₂ cross-link proteomics

293T cells treated with DMSO (vehicle) or Ac₃GlcNDAz-1P(Ac-SATE)₂ from 30 15-cm dishes were harvested, lysed, and quantified as for IP/co-IP. In all, 77 mg of lysate was divided evenly into seven 5-mL centrifuge tubes and rotated with myc antibody overnight at 4 °C. The next day, 50 μL of settled, protein A/G UltraLink resin was washed with IP lysis buffer without EDTA (20 mM Tris-HCl pH 7.4, 1% Triton X-100, 0.1% SDS, 150 mM NaCl) five times and rotated in 600 μL buffer A twice, each 30 min at 4 °C. Ni-NTA purification from the cleared supernatant (1.2 mL total) was performed as above. Resin was washed five times with buffer A and eluted twice with 50 μL elution buffer (250 mM imidazole in 8 M urea/PBS) for 20 min with vigorous shaking at RT. Eluates were separated by SDS-PAGE and stained with colloidal blue. Bands corresponding to NF-L cross-links were excised by hand, digested in-gel as above, and analyzed by MS/MS proteomics by the Duke Proteomics and Metabolomics Shared Resource.

## Differential extraction

Differential extraction experiments were performed essentially as described[68]. Twenty-four hours post-transfection, cells were washed three times with 8 mL of 2 mM MgCl₂/PBS at RT, and agitated in 1 mL of ice-cold low ionic strength (LIS) buffer (10 mM MOPS pH 7, 10 mM MgCl₂, 1 mM EGTA, 0.15% Triton X-100, protease inhibitor cocktail (1:100 in PBS) for 3 min at RT. The supernatant was collected as the LIS fraction. The remaining cells on the dish were incubated in 1 mL of ice-cold high ionic strength (HIS) buffer (10 mM MOPS pH 7, 10 mM MgCl₂, 1% Triton X-100, protein inhibitor cocktail (1:100) and Benzonase nuclease (Novagen, 70746, 1:100) in PBS) for 3 min on ice. Then, 250 μL of ice-cold 5 M NaCl (final 1 M NaCl) was added, cells were resuspended by pipetting and transferred to a clean microfuge tube as the HIS fraction. LIS and HIS fractions were centrifuged at 27,000×g for 15 min at 4 °C, and cleared supernatants were exchanged into IP lysis buffer (20 mM Tris-HCl pH 7.4, 1% Triton X-100, 0.1% SDS, 150 mM NaCl, 1 mM EDTA) using Zeba spin columns. The insoluble pellets from both LIS and HIS tubes were extracted in 200 μL of 8 M urea/PBS on ice for 20 min and probe-sonicated for 50 s at 50% duty cycle. The supernatant was then exchanged to the IP lysis buffer and labeled as the insoluble fraction.

## IFA

In total, 100,000 NEFL⁻/⁻ SH-SY5Y cells seeded in 12-well plates with an 18-mm coverslip on the bottom were transfected with 0.2 μg of each DNA (e.g., NF-L, OGT). Twenty-four hours later, the medium was changed to fresh DMEM/F12. 48 h after transfection, cells were washed twice with PBS, fixed with 4% paraformaldehyde (PFA, MP Biomedicals, 02150146.5 diluted in water) at RT for 15 min, permeabilized with 0.1% Triton X-100/PBS at RT for 10 min, and incubated in blocking buffer (1% BSA/PBS) at RT for 1 h. Coverslips were incubated with the O-GlcNAc (RL2) and NF-L antibodies (1:400 in blocking buffer) overnight at 4 °C, washed three times with PBS, and incubated in Alexa Fluor-conjugated secondary antibody (1:400 in blocking buffer) in the dark at RT for 1 h. Coverslips were washed with PBS twice before mounting in ProLong Diamond anti-fade mounting medium with DAPI (Invitrogen, P36931) onto the microscope slides. For NF-L/NF-M/NF-H co-expression, 200,000 SW13 vim⁻ cells seeded in a 12-well plate were transfected with 0.2 μg NF-L-mCherry DNA, 0.1 μg NF-M-GFP, and 0.05 μg NF-H-3xFLAG using Lipofectamine 3000, processed as above, and stained with the FLAG antibody (1:400 in blocking buffer). For NF-L/INA co-expression, 200,000 SW13 vim⁻ cells seeded in a 12-well plate were transfected with 0.1 μg of each DNA using Lipofectamine 3000, processed as above and stained with the myc and V5 antibodies (1:400 in blocking buffer). Quantification of NF-L morphology in 7–19 cells per condition per biological replicate (for NEFL⁻/⁻ SH-SY5Y cells) or 120-350 cells per condition per biological replicate (for SW13 vim- cells) was performed by a blinded investigator. For rat hippocampal

neurons, cells were fixed in warm 4% PFA/4% sucrose for 8 min, permeabilized with 0.1% Triton X-100/PBS at RT for 5 min, blocked for 1 h in blocking solution (5% goat serum, 0.1% Triton X-100, 0.05% $NaN_3$, 1% BSA/PBS), and stained with Tuj1 antibodies (1:400 in blocking buffer). Following three PBS washes after secondary antibodies, cells were incubated with Hoechst 33342 reagent (ThermoFisher, H3570, 1:2000 in blocking buffer) for 5 min at RT and then washed once with PBS prior to image acquisition. Complete information for all antibodies is provided in Supplementary Table 3.

## Image acquisition (fixed samples)

Cells were imaged on an inverted Zeiss 780 single-point scanning confocal microscope equipped with a fully motorized Zeiss Axio Observer microscope base, Marzhauser linearly encoded stage, diode (405 nm), argon ion (488 nm), double solid-state (561 nm), and helium-neon (633 nm) lasers. Images were acquired at RT using a 63× NA/1.4 oil plan apochromatic oil immersion objective lens. Images were acquired sequentially by frame scanning bidirectionally using the galvanometer-based imaging mode in Zeiss Zen Black Acquisition software and processed using Fiji ImageJ. Detection ranges were 480–500 nm, 490–550 nm, and 650–750 nm.

## Live-cell imaging

On day 6 in vitro, neurons were transfected with 0.3 µg of each DNA using Lipofectamine 3000. Twenty-four hours later, 5–7 axons per condition per biological replicate were imaged live every 5 s over 5 min at 10 ms exposure on a Zeiss ELRYA7 super-resolution microscope equipped with a Pecon environmental chamber at 37 °C, lattice SIM, and Plan-Apochromat 63×/1.4 Oil DIC M27 lens. Time-lapse plus Z imaging was performed with laser excitation at 405 nm, 488 nm, z-step size 110 nm, SIM grating size of 27 µm, and number of SIM phases (13). Laser power was set to <0.5% for each channel to minimize phototoxicity during acquisition. Raw SIM data were processed using SIM Processing in the Zen Software (Zeiss), with "adjusted" setting "normal to baseline cut," followed by "maximum intensity projection." Then, data were processed using IMARIS software with the "surface" function and "track over time" to track distance, displacement, and speed of mitochondria and lysosomes as described[75].

## Statistical analysis

All experimental data presented are representative of multiple independent biological replicates, with replicate numbers indicated in figure legends. All statistical analyses were performed in GraphPad Prism 10 using one-way ANOVA/Tukey's post hoc correction, Student's two-tailed test, or Kruskal–Wallis test/Dunn's post hoc correction, as indicated.

## Reporting summary

Further information on research design is available in the Nature Portfolio Reporting Summary linked to this article.

# Data availability

Raw and processed datasets for O-GlcNAc site-mapping and proteomics experiments generated in this study have been deposited to the MassIVE repository under accession number MSV000091348 and to the ProteomeXchange Consortium via the PRIDE[90] partner repository with the dataset identifier PXD045364. Uncropped immunoblots and raw data are provided in the Source Data file. Unique biological materials described here are available through standard academic material transfer agreements. Source data are provided with this paper.

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

## Acknowledgements

We thank the Light Microscopy Core Facility (LMCF) (Duke) for advice, Dr. So Young Kim (Duke) for assistance with genome engineering and generation of the pSpCas9-GFP-*AAVS1* gRNA construct, Dr. Bin Li (Duke) for help with single-cell sorting, Dr. Chao-Chieh Lin (Duke) for help with qPCR, Noah Linhart (Duke) for help with Ac₃GlcNDAz-1P(Ac-SATE)₂ proteomic sample preparation, Dr. Tetsuya Hirata (Duke) for assistance with immunoblots, and Dr. Natasha Snider (UNC Chapel Hill) for SW13 vim- cells. This work was supported by US National Institutes of Health (NIH) grant R01GM118847 to M.B., NIH grant R01NS111588, and a gift from the Hannah's Hope Foundation to M.B. and J.T.C., a Research Award for Graduate Students from the Ruth K. Broad Biomedical Research Foundation to D.T.H., a Hannah Gray Fellowship from the Howard Hughes Medical Institute and support from the Duke Science and Technology Scholars program to C.S.E., NIH grant S10D028703 to the Duke LMCF for the Zeiss ELYRA7 microscope, and NIH Alzheimer's Disease Research Center grant P30AG072958 to the Duke Bryan Brain Bank and Biorepository.

## Author contributions

M.B. and D.T.H. conceived and supervised the project. D.T.H., K.N.T., A.J.W., S.K.K., D.L., and J.H. performed the experiments. J.R.G. and C.S.E. prepared the rat hippocampal neurons and consulted on organelle motility assays and data analysis. E.J.S. performed the MS-identification of human NF-L O-GlcNAc sites and Ac₃GlcNDAz-1P(Ac-SATE)₂ proteomics. J.T.C. consulted on the strategic directions of all experiments. M.B. and D.T.H. wrote and all authors reviewed the manuscript.

## Competing interests

The authors declare no competing interests.
