## [Peer Review File · Nature Communications]

O-GlcNAcylation regulates neurofilament-light assembly and function and is perturbed by Charcot-Marie-Tooth disease mutationsREVIEWER COMMENTS

Reviewer #1 (Remarks to the Author):

NCOMMS-23-09578- 417238

The authors submitted the manuscript entitled: O-GlcNAcylation regulates neurofilament-light assembly and function and is perturbed by Charcot-Marie-Tooth disease mutations. The manuscript is of a high scientific value since O-GlcNAc modifications have been consistently found in proteomic analysis of neurofilament proteins without finding a clear function to these post-translational modifications. The authors identified O-GlcNAc sites in NFL and show that they regulate NF solubility. They also identify in CMT2E-causing variants abnormal glycosylation which stresses the important role of this PTM for neurofilament biology and CMT2E pathogenesis. The experiments are well carried out using top techniques and the demonstration of the role of OGT very clear. In addition, the authors suggest that this PTM on NFL may work as a sensing mechanism for nutrients like glucose and glutamine and opens the question on the role of this sensing in particular in maintaining assembled neurofilaments. The manuscript is particularly well-constructed and the use of several models impressive. Congratulations for this very rigorous work. O-GlcNAc modifications are responsible of a slight switch of NFL from the urea fraction to the 'soluble' fraction hence the conclusion that O-GlcNAc regulates assembly into higher order structures. Because NF assembly has been shown to be regulated by phosphorylation and dephosphorylation, it is not excluded that the O-GlcNAc acts indirectly through phosphorylation events. The authors will need to investigate the impact of O-GlcNAc on phosphorylation levels of NFL using phosphospecific antibodies. However, I concur with their statement that dissecting the mechanism and finding the potential crosstalk between phosphorylation and O-GlcNAc signalisations will require future studies.

Major comment: More evidence will reinforce the claim of the authors: i.e. O-GlcNAcs are involved in assembly of NFL.

The effect of O-GlcNAc on NF network deserves a deeper characterisation so it can be compared with the effect of CMT2E variants published in the literature. The interaction with INA is often observed in immature neuronal systems like SHSY cells and do not reflect the complexity of neurofilament assembly and composition. Neurofilaments are obligate hetero polymers made of NFL-NFM-NFH with either INA or peripherin (ratio of 4:2:1:1); NFL being the core protein for filamentous formation. If expressing the 3 NF proteins at the same time in SW13 remains difficult, the authors may take advantage of the models used in figure 3 (hippocampal neurons) and of the SW13- model to tight their observation with the abundant litterature (see Beaulieu et al, Sasaki et al, Zhai et al, McLean et al., Gentil et al. the work from Liem and Browns' groups etc...). They may express NFL4A or NFL5A and co-immunolabel with either NFM or NFH to characterise the NF network hippocampal neurons. Another possibility is to take advantage of the SW13 model and co-transfect plasmids encoding NFL4A or 5A together with plasmids encoding NFM or NFH, and to analyse the effect of NFL on the NF network. In conclusion, a better analysis of the effect of oGlcNAc modifications in NFL on the NFs network will be needed.

Minor comment: The demonstration of the direct involvement of O-GlcNAc in NFL assembly could be demonstrated by in vitro assembly of recombinant NFLwt and NFL4A and observation of filamentous structure following negative staining and electron microscopy imaging.

Reviewer #2 (Remarks to the Author):

I was asked to provide a review on the UV-crosslinking parts of the manuscript. The authors use a well-known strategy originally introduced by Kohler and colleagues to cross-link O-GlcNAc to interaction partners, using the diazirine-containing sugar GlcNDaz.

The crosslinking experiments in Fig. 5 are convincing and largely support the authors' hypotheses. NF-L clearly engages in high Mw complexes. At this stage, it should be noted that 1) it is normal

for GlcNDAz to be incorporated substoichiometrically, leading to incomplete crosslinking; and 2) that some crosslinks occur from just protein proximity in the presence of harsh UV light (and thereby without a diazirine, as shown in Fig. 5e, right panel). Nevertheless, the data in the other panels are clear.

I do not agree with the statement that the data in Fig. 5d suggest homotypic interactions. This could be proved if a pull-down was performed on either the myc- or V5-tagged NF-L with detection of the respective other tag. As it stands, while it is likely that they interact, these data only suggest that both myc- and V5-tagged NF-L are found in complexes of the same Mw.

Other points:

- please try to make Western Blots bigger - the data are barely readable at times.
- can the authors please state the number of independent replicates for the experiments in 5b-e?
- can the authors please use a different name for the synthetic compound rather than "GlcNDAz", which is already assigned to the free sugar? They could just use the full compound name.

Reviewer #3 (Remarks to the Author):

In this paper, the authors study the effect of O-GlcNAcylation on neurofilament assembly, looking particularly at NF-L (neurofilament light). Dr. Boyce's laboratory is an expert on protein glycosylation, and consequently, the first set of experiments are thoroughly and expertly performed. There are some small details that could be clarified as delineated below. NF-L is known to be one of the genes mutated in Charcot-Marie-Tooth (CMT) disease. The authors then perform a series of experiment to determine whether CMT-causative NF-L mutants exhibit perturbed O-GlcNAc levels and resist the effects of O-GlcNAcylation on NF assembly state. Although there is an effect, different mutations behave differently (sometimes in opposite directions, thereby putting into question that O-GlcNAcylation contributes to CMT).

Starting with the latter point first, the authors show in Fig. 6, how the different mutations effect O-GlcNAc content vs. wild type. The results are completely dependent on the mutations. A number of mutations in the head domain of NFL result in very low levels of O-GlcNAc content, whereas the mutations in the rod domain are either not affected, or in the case of N98S and E396K are increased by as much as 75% for N98S. However, the results in panel f and g show that there is no correlation between the content of O-GlcNAc and either the percent of cells with full-length NF-L or the crosslink/uncrosslinked ratio, although both are changed in a number of the mutants. Thus, it seems unlikely that the level of O-GlcNAcylation contributes to CMT.

As noted, the effect of O-GlcNAcylation on NF-L assembly is quite interesting and seem to be well-documented. There are a few additional questions about these data that the authors should clarify.

1. The IP experiments in Fig. 1 are done by solubilizing NF-L in a buffer containing 1% Triton X-100 and 0.1% SDS. Does this solubilize all the NF-L? In Fig. 2, the authors describe NF-L with different levels of solubility in low ionic strength, high ionic strength and urea containing buffers. I was wondering if the authors determined how much NF-L was left after solubilizing in the Triton X-100/0.1% SDS buffer, and whether this makes a difference in their interpretation,
2. The authors list three different antibodies to O-GlcNAc in their table of antibodies, but don't specify when they use each of these antibodies, or whether using different antibodies matter (it may not, but they should state that)
3. I am curious as to what happens with vimentin. NF-L can co-assemble with vimentin, yet it does not show up as one of the possible partners in Table 5. Is there not much vimentin in these cells? Have they looked? I am also not sure how significant the results with INA in Figure 5 are in terms of mechanism of NF-L assembly
4. The authors start by using 293T cells but then switch to SH-SY5 cells. The latter have NF-L knocked out, but are the results dependent on the absence of endogenous NF-L in these cells? What other intermediate filaments are in the NF-L-/- SH-SY5 cells? Do they matter?

Overall, the authors present data on a new post-translational modification of NF-L and show that

this modification has an effect of assembly of NF-L, as well as mitochondrial transport. As such there is new and interesting data in this paper. Their attempt to relate this modification to CMT-disease is not so convincing however.

Thank you for the reviews of our manuscript, “O-GlcNAcylation regulates neurofilament-light assembly and function and is perturbed by Charcot-Marie-Tooth disease mutations,” NCOMMS-23-09578-417238. We appreciate the reviewers’ positive comments about the novelty, rigor, clarity, and significance of our results and conclusions, and we are grateful for the opportunity to improve our manuscript further through responses to their constructive criticisms and questions. Below we provide a point-by-point reply to all reviewers’ comments. We have also extensively revised the manuscript according to the comments from the reviewers and editorial team (with revisions highlighted in yellow, as instructed) and have provided all additional documents and information in our resubmission package. We look forward to the re-review.

Reviewer 1

Comment 1: The authors submitted the manuscript entitled: O-GlcNAcylation regulates neurofilament-light assembly and function and is perturbed by Charcot-Marie-Tooth disease mutations. The manuscript is of a high scientific value since OGlcNac modifications have been consistently found in proteomic analysis of neurofilament proteins without finding a clear function to these post-translational modifications. The authors identified O-GlcNAc sites in NFL and show that they regulate NF solubility. They also identify in CMT2E-causing variants abnormal glycosylation which stresses the important role of this PTM for neurofilament biology and CMT2E pathogenesis. The experiments are well carried out using top techniques and the demonstration of the role of OGT very clear. In addition, the authors suggest that this PTM on NFL may work as a sensing mechanism for nutrients like glucose and glutamine and opens the question on the role of this sensing in particular in maintaining assembled neurofilaments. The manuscript is particularly well-constructed and the use of several models impressive. Congratulations for this very rigorous work.

Reply 1: We thank the reviewer for these positive remarks and are glad for the chance to strengthen our manuscript further through responses to the comments below.

Comment 2: OGlcNac modifications are responsible of a slight switch of NFL from the urea fraction to the ‘soluble’ fraction hence the conclusion that OGlcNac regulates assembly into higher order structures. Because NF assembly has been shown to be regulated by phosphorylation and dephosphorylation, it is not excluded that the OGlcNac acts indirectly through phosphorylation events. The authors will need to investigate the impact of OGlcNac on phosphorylation levels of NFL using phosphospecific antibodies. However, I concur with their statement that dissecting the mechanism and finding the potential crosstalk between phosphorylation and OGlcNac signalisations will require future studies.

Reply 2: We agree with both parts of the reviewer’s comment: Though a comprehensive characterization of the interplay between NF-L glycosylation and phosphorylation is beyond the scope of the current work, additional data and discussion of the relationship between these post-translational modifications would improve our manuscript. To address this point, we have performed new experiments with phospho-specific antibodies, as requested (Supplementary

Figure 1E-G in the resubmission). We now demonstrate that wild type NF-L phosphorylation was reproducibly reduced by OGT co-expression (Supplementary Figure 1F). Conversely, treatment with the PP1/PP2A phosphatase inhibitor calyculin A elevated NF-L phosphorylation and diminished O-GlcNAcylation (Supplementary Figure 1G). Finally, mutations at NF-L O-GlcNAcylation sites modestly increased baseline phosphorylation, relative to wild type (Supplementary Figure 1E). Taken together, these new results suggest that NF-L phosphorylation and glycosylation may be reciprocal modifications under at least some conditions. However, none of the NF-L O-GlcNAc sites that we characterized is a validated phosphorylation site, so this putative reciprocal signaling likely involves as-yet unidentified phosphorylation sites. Therefore, as the reviewer anticipates, we believe a thorough investigation of this signaling is a promising direction for future studies. We have included these new data and additional discussion in the revised manuscript.

Comment 3: Major comment: More evidence will reinforce the claim of the authors: i.e. OGlcNAcs are involved in assembly of NFL. The effect of OGlcNAc on NF network deserves a deeper characterisation so it can be compared with the effect of CMT2E variants published in the literature. The interaction with INA is often observed in immature neuronal systems like SHSY cells and do not reflect the complexity of neurofilament assembly and composition. Neurofilaments are obligate hetero polymers made of NFL-NFM-NFH with either INA or peripherin (ratio of 4:2:1:1); NFL being the core protein for filamentous formation. If expressing the 3 NF proteins at the same time in SW13 remains difficult, the authors may take advantage of the models used in figure 3 (hippocampal neurons) and of the SW13- model to tight their observation with the abundant litterature (see Beaulieu et al, Sasaki et al, Zhai et al, McLean et al., Gentil et al. the work from Liem and Browns' groups etc...). They may express NFL4A or NFL5A and co-immunolabel with either NFM or NFH to characterise the NF network hippocampal neurons. Another possibility is to take advantage of the SW13 model and co-transfect plasmids encoding NFL4A or 5A together with plasmids encoding NFM or NFH, and to analyse the effect of NFL on the NF network. In conclusion, a better analysis of the effect of oGlcNac modifications in NFL on the NFs network will be needed.

Reply 3: We agree with the reviewer that more studies on the effects of NF-L O-GlcNAcylation on the NF network, as outlined above, would bolster our model. To address this point, we co-expressed wild type NF-L or the NF-L^{4A} glycosite mutant with NF-M or NF-H in SW13 vim⁻ cells at a physiological 4:2:1 ratio, as suggested by the reviewer, and analyzed filament morphology by immunofluorescence assay (IFA) (Supplementary Figure 2K-L). In these experiments, NF-L^{4A} demonstrated reduced co-assembly with NF-M or NF-H, relative to wild type NF-L (Supplementary Figure 2K-L). These observations are consistent with our model that O-GlcNAcylation tunes the assembly of NF-L into full-length filaments with itself and with other neurofilament proteins. In the revised manuscript, we have included these new results as well as additional discussion of this point and the appropriate literature mentioned by the reviewer. As the reviewer notes, it will be interesting for future studies to determine how the effects of NF-L O-GlcNAcylation on IF morphology compare to the impact of CMT2E/F variants under physiological and stress conditions.

Comment 4: *Minor comment: The demonstration of the direct involvement of OGlcNac in NFL assembly could be demonstrated by in vitro assembly of recombinant NFLwt and NFL4A and observation of filamentous structure following negative staining and electron microscopy imaging.*

Reply 4: We agree with the reviewer that an appropriate *in vitro* assembly assay would be a powerful tool for studying the biochemical effects of NF-L O-GlcNAcylation on IF morphology and regulation. No suitable system currently exists, so we endeavored to establish one for our revised manuscript. We succeeded in expressing GST-tagged human wild type NF-L and NF-L^{4A} in *E. coli* (Reviewer Figure 1A, below). Although the solubility of both proteins was poor, as expected, we were able to purify workable amounts of soluble NF-L fusion proteins via glutathione affinity chromatography (Reviewer Figure 1A). To address the reviewer's question, we would next need: 1) a means of O-GlcNAcylation of recombinant-purified NF-L (*E. coli* lack endogenous O-GlcNAcylation) and 2) a quantitative assay of NF-L assembly. To address the first requirement, we subjected the GST-NF-L proteins to *in vitro* glycosylation reactions with recombinant-purified human OGT, a system that we have used previously (PMID 29784830) (Reviewer Figure 1B). Unfortunately, while the positive control substrate, casein kinase 2 (CK2), was readily O-GlcNAcylation by OGT, we detected no evidence of modification of wild type or mutant NF-L (Reviewer Figure 1B). This result may indicate that OGT requires specific, unidentified cofactor proteins to glycosylate NF-L, as has been observed for other validated OGT substrates (PMIDs 18353774, 22883232, 16505006). Furthermore, we note that, even if a robust *in vitro* NF-L glycosylation system were developed, significant effort (e.g., extensive mass spectrometry characterization) would be required to confirm that the O-GlcNAcylation occurs at the same physiological sites observed *in vivo*, a considerable additional technical hurdle. For the second requirement, we attempted to recapitulate a spectroscopic assay previously reported to quantify the *in vitro* assembly of GST-tagged rat NF-L purified from *E. coli* (PMID 21339697). Unfortunately, we did not detect any evidence of assembly by either wild type NF-L or NF-L^{4A} under these conditions (Reviewer Figure 1C), perhaps indicating that the assay is sensitive to small differences in GST fusion linkers and/or is suitable for rodent but not human NF-L. Based on these negative results, we believe that establishing *in vitro* human NF-L O-GlcNAcylation and assembly systems may require a sizeable amount of optimization and additional, labor-intensive assays (e.g., negative-stain electron microscopy). These considerations unfortunately make these objectives impractical for the current work, but they will remain important goals for future studies.

Reviewer 2

Comment 1: *I was asked to provide a review on the UV-crosslinking parts of the manuscript. The authors use a well-known strategy originally introduced by Kohler and colleagues to cross-link O-GlcNAc to interaction partners, using the diazirine-containing sugar GlcNDAz.*

The crosslinking experiments in Fig. 5 are convincing and largely support the authors' hypotheses. NF-L clearly engages in high Mw complexes. At this stage, it should be noted that 1) it is normal for GlcNDAz to be incorporated substoichiometrically, leading to incomplete crosslinking; and 2)

that some crosslinks occur from just protein proximity in the presence of harsh UV light (and thereby without a diazirine, as shown in Fig. 5e, right panel). Nevertheless, the data in the other panels are clear.

Reply 1: We thank the reviewer for his or her positive assessment of our Ac₃GlcNDAz-1P(Ac-SATE)₂ crosslinking data. We agree that it is important to note that O-GlcNDAz is sub-stoichiometric (even compared to O-GlcNAc) and that background crosslinking due to UV alone is commonly observed (e.g., PMIDs 22411826, 29513221). We have added these points to the revised manuscript.

Comment 2: *I do not agree with the statement that the data in Fig. 5d suggest homotypic interactions. This could be proved if a pull-down was performed on either the myc- or V5-tagged NF-L with detection of the respective other tag. As it stands, while it is likely that they interact, these data only suggest that both myc- and V5-tagged NF-L are found in complexes of the same Mw.*

Reply 2: We agree with the reviewer that a reciprocal IP/immunoblot between myc- and V5-tagged NF-L in the Ac₃GlcNDAz-1P(Ac-SATE)₂-treated, crosslinked samples is required to demonstrate homotypic NF-L/NF-L interactions. In fact, this is the experiment depicted in Figure 5D, which shows myc and V5 immunoblots of myc-IP-ed samples. We apologize that this aspect of the experimental design was not clear in the original submission. In the revised manuscript, we have edited the Results section text and the image and legend for Figure 5D to clarify this point.

Comment 3: *Other points: - please try to make Western Blots bigger - the data are barely readable at times.*

Reply 3: We apologize for the difficulty with the size of the immunoblot data. In the revised manuscript, we have increased the size of all blots to the maximum extent possible while still adhering to journal figure formatting requirements.

Comment 4: *- can the authors please state the number of independent replicates for the experiments in 5b-e?*

Reply 4: We apologize that the information on the number of biological replicates for Figure 5B-E was not clear. General information on replicate numbers was included in the Materials and Methods section of the original manuscript. In the revised submission, we have also listed the number of biological replicates in every pertinent figure legend, including for Figure 5.

Comment 5: *- can the authors please use a different name for the synthetic compound rather than "GlcNDAz", which is already assigned to the free sugar? They could just use the full compound name.*

Reply 5: We agree with the reviewer that it would be more correct to refer to the precursor photocrosslinking sugar reagent by its complete abbreviation, Ac₃GlcNDAz-1P(Ac-SATE)₂, rather than the “GlcNDAz” shorthand used in the original submission, which more properly refers to the free sugar. We have made this revision throughout the resubmitted manuscript and explained the abbreviation upon first usage.

Reviewer 3

Comment 1: *In this paper, the authors study the effect of O-GlcNAcylation on neurofilament assembly, looking particularly at NF-L (neurofilament light). Dr. Boyce’s laboratory is an expert on protein glycosylation, and consequently, the first set of experiments are thoroughly and expertly performed. There are some small details that could be clarified as delineated below. NF-L is known to be one of the genes mutated in Charcot-Marie-Tooth (CMT) disease. The authors then perform a series of experiment to determine whether CMT-causative NF-L mutants exhibit perturbed O-GlcNAc levels and resist the effects of O-GlcNAcylation on NF assembly state. Although there is an effect, different mutations behave differently (sometimes in opposite directions, thereby putting into question that O-GlcNAcylation contributes to CMT.*

Reply 1: We thank the reviewer for the positive remarks about our lab and the experiments in the current manuscript. We agree that the heterogeneous phenotypes we observed among the Charcot-Marie-Tooth (CMT)-causative mutants indicate that the effects on NF-L O-GlcNAcylation may vary by mutation in both experimental systems and in CMT patients. Indeed, this heterogeneity of biochemical, cellular, and clinical phenotypes is a well-known (albeit still mysterious) feature of CMT2E/F, the disease subtypes caused by *NEFL* mutations. As the reviewer implies, these facts call for caution in interpreting the biochemical and cellular impacts of O-GlcNAcylation on CMT-causative NF-L mutants. We have extensively revised the Discussion section to emphasize these important points and to use appropriately conservative language in describing our data, hypotheses, and conclusions regarding CMT mutants.

Comment 2: *Starting with the latter point first, the authors show in Fig. 6, how the different mutations effect O-GlcNAc content vs. wild type. The results are completely dependent on the mutations. A number of mutations in the head domain of NFL result in very low levels of O-GlcNAc content, whereas the mutations in the rod domain are either not affected, or in the case of N98S and E396K are increased by as much as 75% for N98S. However, the results in panel f and g show that there is no correlation between the content of O-GlcNAc and either the percent of cells with full-length NF-L or the crosslink/uncrosslinked ratio, although both are changed in a number of the mutants. Thus, it seems unlikely that the level of O-GlcNAcylation contributes to CMT.*

Reply 2: We agree with the reviewer that the functional impacts of O-GlcNAcylation on NF-L CMT mutants are heterogeneous, and we have not yet fully characterized the hypothesized functional relationships between NF-L O-GlcNAcylation and each CMT-causative mutant. We have extensively revised the Discussion section of the manuscript to better highlight this important general point. On the other hand, some trends do emerge from our data that suggest testable hypotheses for future studies. For example, we noted that CMT-causative mutations near known

glycosites uniformly reduced NF-L O-GlcNAcylation (Figures 6A-B) and demonstrated aggregation and loss of lower-order, soluble NF-L assembly states (Figures 6C-F). Importantly, these results are consistent with our data on wild type NF-L, where increased O-GlcNAcylation drives the protein to lower-order assembly states (Figure 2). Taken together, these observations suggest that CMT mutations near glycosites may impair NF-L function in part by inhibiting O-GlcNAcylation, though future studies with more mutants and additional experimental systems will be necessary to rigorously test this hypothesis. Similarly, the functional interactions between other CMT-causative mutations and NF-L O-GlcNAcylation remain to be dissected, a goal that we believe to be important and interesting, yet beyond the scope of this manuscript. We have revised the portion of the Discussion section that considers our CMT mutant data to elaborate on these caveats and future objectives.

Comment 3: As noted, the effect of O-GlcNAcylation on NF-L assembly is quite interesting and seem to be well-documented.

Reply 3: We appreciate the reviewer's positive assessment of the interest and evidentiary support of our conclusions.

Comment 4: There are a few additional questions about these data that the authors should clarify. 1. The IP experiments in Fig. 1 are done by solubilizing NF-L in a buffer containing 1% Triton X-100 and 0.1% SDS. Does this solubilize all the NF-L? In Fig. 2, the authors describe NF-L with different levels of solubility in low ionic strength, high ionic strength and urea containing buffers. I was wondering if the authors determined how much NF-L was left after solubilizing in the Triton X-100/0.1% SDS buffer, and whether this makes a difference in their interpretation,

Reply 4: We agree with the reviewer that the biochemical aspects of our sample preparations could hypothetically impact our conclusions. For the biochemical experiments performed in HEK293T cells mentioned by the reviewer, extracts were made directly in 1% Triton X-100/0.1% SDS buffer. Under these conditions, trace amounts of NF-L remain in the insoluble pellet, as determined by subsequent extraction of the pellet with 8 M urea and analysis by immunoblot. We have revised the Results section to reflect this consideration. Importantly, no data interpretation or conclusions are affected by these observations because of the low proportion of residual insoluble NF-L under these conditions and because our biochemical experiments are complemented by independent methods, such as IFA, throughout.

Comment 5: 2. The authors list three different antibodies to O-GlcNAc in their table of antibodies, but don't specify when they use each of these antibodies, or whether using different antibodies matter (it may not, but they should state that)

Reply 5: We apologize for this ambiguity and agree with the reviewer that clearer information on the O-GlcNAc antibodies is needed. In the revised manuscript, we have specified the particular anti-O-GlcNAc monoclonal antibody used in each experiment in the corresponding figure image and/or legend. Several different anti-O-GlcNAc monoclonal antibodies readily recognize glycosylation on NF-L, and we have used them essentially interchangeably in the current work.

Comment 6: 3. I am curious as to what happens with vimentin. NF-L can co-assemble with vimentin, yet it does not show up as one of the possible partners in Table 5. Is there not much vimentin in these cells? Have they looked? I am also not sure how significant the results with INA in Figure 5 are in terms of mechanism of NF-L assembly

Reply 6: As the reviewer notes, we did not detect vimentin in our proteomic analyses of Ac₃GlcNDAz-1P(Ac-SATE)₂-treated, crosslinked NF-L samples (Table 5), although HEK293T cells indeed express endogenous vimentin and can exhibit homopolymeric vimentin filaments (e.g., PMID 29513221). To address the reviewer's question via another assay, we performed O-GlcNDAz crosslinking on NF-L-myc-6xHis-expressing cells, purified the crosslinked complexes by tandem myc immunoprecipitation and Ni-NTA, and analyzed them by immunoblot (Reviewer Figure 2, below). Consistent with our mass spectrometry proteomics results, endogenous vimentin was not detected in the crosslinks (Reviewer Figure 2). These observations indicate that vimentin and NF-L do not engage in O-GlcNAc-mediated protein-protein interactions in this system, though some degree of vimentin/NF-L co-assembly remains theoretically possible.

Regarding the significance of our INA results, our experiments revealed a previously unknown, O-GlcNAc-mediated interaction between INA and NF-L and demonstrated that site-specific glycosylation of NF-L promotes this interaction (Figure 5). As the reviewer's comment suggests, the *in vivo* significance of these observations remains an open question for future work. The Nixon lab and other groups have shown that INA co-assembles with the triplet proteins (including NF-L) in NFs in adult neurons (e.g., PMIDs 17005864, 2413040), implying that INA/NF-L interactions and their post-translational regulation may be functionally important *in vivo*. It will be important to test these possibilities in physiological systems in future work. We have significantly revised the portion of the Discussion section on our INA results to reflect these considerations.

Comment 7: 4. The authors start by using 293T cells but then switch to SH-SY5Y cells. The latter have NF-L knocked out, but are the results dependent on the absence of endogenous NF-L in these cells? What other intermediate filaments are in the NF-L^{-/-} SH-SY5Y cells? Do they matter?

Reply 7: We agree with the reviewer that characterizing other neurofilament proteins in our control and *NEFL*^{-/-} SH-SY5Y systems could be helpful for the interpretation of our results. Importantly, we have previously found that phenotypes of unglycosylatable mutants of intermediate filament proteins can be suppressed in the presence of the wild type protein (PMID 29513221), motivating us to create *NEFL*^{-/-} SH-SY5Y cells for this study. To address the reviewer's point about this new model system, we performed IFA on control and *NEFL*^{-/-} SH-SY5Y (now provided as Supplementary Figure 2F). Endogenous NF-H and INA were readily observed in both cell systems (Supplementary Figure 2F), whereas NF-M was not detectable in either (not shown). We did not detect any major, genotype-dependent differences in the distribution or morphology of NF-H or INA (Supplementary Figure 2F). These results support our conclusion that *NEFL*^{-/-} SH-SY5Y cells provide an appropriate null background for examining the functional effects of NF-L mutations and do not notably deviate from control cells with respect to other neurofilament proteins tested.

Comment 8: *Overall, the authors present data on a new post-translational modification of NF-L and show that this modification has an effect of assembly of NF-L, as well as mitochondrial transport. As such there is new and interesting data in this paper. Their attempt to relate this modification to CMT-disease is not so convincing however.*

Reply 8: We thank the reviewer for the positive comments on the novelty and significance of our data on NF-L O-GlcNAcylation. We also understand the reviewer's concern regarding the implications of our results for CMT disease. As mentioned in Replies 1 and 2 above, we completely agree that further studies will be needed to dissect the potential functional interplay between CMT-causative mutations and NF-L O-GlcNAcylation in experimental animal models and in CMT patient samples. It is important to note that these studies may be complicated by the well-documented heterogeneity in biochemical, cellular, and clinical phenotypes among CMT-causative NF-L mutants (PMIDs 33993654), a phenomenon also reflected in our own experiments (Figure 6). Understanding these aspects of NF-L O-GlcNAcylation, its regulation, and its pathophysiological significance will be key goals of future work. We have extensively revised the Discussion section to better highlight these caveats and to stress the need for additional studies of NF-L O-GlcNAcylation in CMT models.

Reviewer Figure 1 **a** GST-tagged wild type NF-L or the NF-L^{4A} glycosite mutant (or empty vector, control) was expressed in *E. coli*, purified by glutathione affinity chromatography, and analyzed by SDS-PAGE and colloidal blue stain. Arrow indicates the ~105 kDa GST fusion proteins, which are modestly soluble, as expected. **b** GST-tagged wild type NF-L or NF-L^{4A} was subjected to *in vitro* O-GlcNAcylation with recombinant-purified human OGT essentially as described (PMID 29784830) and analyzed by immunoblot. GST and CK2 are negative and positive controls, respectively. No O-GlcNAc signal on wild type or mutant GST-NF-L protein was detected by either of two anti-O-GlcNAc monoclonal antibodies. **c** GST-tagged wild type NF-L or NF-L^{4A} (or GST only, control) was subjected to *in vitro* filament assembly conditions and monitored at OD₃₅₀ on a spectrophotometer, as described (PMID 21339697). No significant change in OD₃₅₀ was observed in any reaction, indicating little or no filament assembly.

Reviewer Figure 2 293T cells were transfected with NF-L-myc-6xHis \pm 100 μ M Ac₃GlcNDAz-1P(Ac-SATE)₂ for 48 hrs and subjected to UV crosslinking. NF-L-myc-6xHis was affinity-purified by tandem myc immunoprecipitation/Ni-NTA and analyzed by immunoblot. Dark exposures of relevant sections of the myc and vimentin blots are shown to illustrate NF-L crosslinking and the absence of detectable vimentin in the crosslinks.

REVIEWERS' COMMENTS

Reviewer #1 (Remarks to the Author):

The authors have answered all the questions. I have no further comment. Congratulations for this excellent piece of work.

Reviewer #2 (Remarks to the Author):

Thank you for addressing my comments. Sorry for misreading figure 5D!

Reviewer #3 (Remarks to the Author):

The reviewers have appropriately responded to my previous comments, especially about modifying the statements about possible CMT mechanism.